# Cooperative Ternary Assemblies Involving Anion–π/π–π/Anion–π Assemblies and Unconventional Cl···Cl Interactions in Cu(II) Coordination Compounds: Experimental and Theoretical Studies

**Pinku Sarma** [1], **Rosa M. Gomila** [2], **Antonio Frontera** [2,*], **Miquel Barcelo-Oliver** [2] **and Manjit K. Bhattacharyya** [1,*]

[1] Department of Chemistry, Cotton University, Guwahati 781001, India
[2] Departament de Química, Universitat de les Illes Balears, Crta de Valldemossa km 7.7, 07122 Palma de Mallorca, Baleares, Spain
\* Correspondence: toni.frontera@uib.es (A.F.); manjit.bhattacharyya@cottonuniversity.ac.in (M.K.B.)

**Abstract:** Two coordination compounds of Cu(II), namely, [Cu (phen)$_2$Cl](NO$_3$)·H$_2$O (compound **1**) and [Cu$_2$(μ-Cl$_2$)Cl$_2$(Hdmpz)$_4$] (compound **2**), where phen = 1,10-phenanthroline and Hdmpz = 3,5-dimethylpyrazole, were synthesized at room temperature and characterized using elemental analysis, TGA, spectroscopic techniques (FT-IR and electronic) and single-crystal X-ray diffraction studies. The cooperative anion–π/π–π/anion–π assemblies involving the coordinated phen, along with the uncoordinated nitrate moieties, played pivotal roles in the stabilization of the crystal structure of compound **1**. Unconventional type I Cl···Cl interactions involving the coordinated Cl atoms provided reinforcement to the crystal structure of compound **2**. We theoretically explored the supramolecular assemblies observed in the crystal structures of compounds **1** and **2** using DFT calculations, MEP surface analysis and combined NCI plot/QTAIM computational tools. Theoretical analysis revealed that the antiparallel π-stacking interactions in compound **1** and the N–H···Cl H-bonds in compound **2** were the strong structure-guiding non-covalent synthons which stabilized the compounds. In the anion–π/π–π/anion–π assembly observed in compound **1**, the anion–π interaction reinforced the π-stacking by reducing the electrostatic repulsion between the metal-coordinated electron-deficient phen rings.

**Keywords:** coordination compound; cooperative assemblies; Cl···Cl interaction; DFT calculations; NCI plot/QTAIM analysis





## 1. Introduction

The design, synthesis and development of coordination complexes of transition metals have attracted significant attention from researchers not only due to their myriad potential applications in various fields but also due to their intriguing structural topologies [1–4]. Organization of the molecular building blocks via a self-assembly process plays a decisive role in the structural topology and network architectures of coordination compounds [5,6]. However, it is challenging to develop network architectures of the desired dimensionalities as various synthetic factors such as the coordination behavior of the metal centers, organic moieties as ligands, pH, temperature, etc. can effectively influence the design and synthesis of the desired structural topologies [7–9]. The choice of solvent also plays a crucial role in the design of coordination compounds with fascinating network architectures, as solvent molecules can coordinate with the metal centers and/or are present in the crystal lattices, facilitating diverse structural topologies [10,11].

Supramolecular chemistry deals with the non-covalent interactions that play decisive roles in various processes such as in synthetic chemistry, catalysis, the development of

pharmaceutical agents, molecular biology, etc. [12–14]. Myriad attempts have been made in recent times to quantify the self-assembly of non-covalent interactions in coordination compounds [15–19]. Among the various non-covalent interactions, hydrogen-bonding interactions are considered as the most common non-covalent interaction [20]. Coordination compounds involving aromatic ligands are of particular interest in supramolecular chemistry as the aromatic rings can facilitate the $\pi$-stacking interactions thatplay a crucial role in the structural topologies of metal–organic compounds [21,22]. Aromatic $\pi$-stacking interactions are also involved in the self-assembly of bio-molecules, namely, the stabilization of DNA's helix structure [23,24], the interactions of drugs/compounds with proteins, etc. [25]. Non-covalent interactions involving anions with electron-deficient aromatics, namely, anion–$\pi$ interactions, also contribute to the solid state stabilities of supramolecular architectures [26]. The cooperative interplay of non-covalent interactions has also received remarkable emphasis from a crystal engineering viewpoint [27,28]. Unusual non-covalent interactions such as dihydrogen bonding, halogen bonding, etc. also play critical roles in the organization of the structures of the compounds in asolid state, which can also affect the properties of the compounds [29–33]. Unconventional Cl$\cdots$Cl contacts have also received significant attention in supramolecular chemistry, which can also be considered as donor–acceptor interactions [34]. Usually, the intermolecular C-X1$\cdots$X2-C interactions (where X = F, Cl, Br or I), depending on the angles, are of the following two types: $\theta1 = \angle$C-X1$\cdots$X2 and $\theta2 = \angle$X1$\cdots$X2-C. The interactions with $\theta1 = \theta2$ are called type I, whereas the condition $\theta1 \neq \theta2$ belongs to the interactions of type II [35]. Various research groups have already established the donor–acceptor behavior of halogen$\cdots$halogen interactions using computational studies [36,37]. The uncoordinated anions present in the crystal lattice primarilycompensate the overall positive charges of the complex cationic moieties; however, they can also facilitate unusual non-covalent interactions [38,39].

1,10-Phenanthrline (phen), a classic bidentate chelating N-donor moiety, is capable of interacting with various biological systems both as a free ligand and as a coordinated complex to the metal centers [40]. Phen is capable of forming stable coordination complexes with most of the transition metal centers, thereby holding a peculiar place as starting building block in coordination chemistry [41–46]. The presence of electron-deficient aromatic systems in phen makes it an excellent electron acceptor capable of stabilizing metal complexes via various unusual non-covalent interactions [44]. Pyrazole and substituted derivatives and aromatic heterocycles have been extensively used to construct transition metal compounds [47–49]. Both phen and Hdmpz, which are N-donor heterocycles, can be effectively used as building blocks to develop coordination compounds of the desired dimensionalities, with potential applications [50–53].

In the present study, we have reported the synthesis and crystal structures of two coordination compounds of Cu(II), namely, [Cu(phen)$_2$Cl](NO$_3$)·H$_2$O (compound **1**) and [Cu$_2$($\mu$-Cl$_2$)Cl$_2$(Hdmpz)$_4$] (compound **2**), where phen = 1,10-phenanthroline and Hdmpz = 3,5-dimethylpyrazole). We have further characterized the compounds using various spectroscopic and analytical techniques such as elemental analysis, TGA and spectroscopic (FT-IR and electronic) techniques. The uncoordinated nitrate anion and water molecule, along with the aromatic $\pi$-stacking involving the coordinated phen moieties, stabilized the crystal structure of compound **1**. Unusual type I Cl$\cdots$Cl contacts involving the coordinated Cl atoms provided stability to the crystal structure of compound **2**. The energetic features of the unconventional non-covalent interactions observed in the compounds have been explored using theoretical studies, with special attention devoted to the anion–$\pi$/($\pi$–$\pi$)/anion–$\pi$ ternary assemblies in compound 1 and the N–H$\cdots$Cl H bonds in compound **2**. The supramolecular assemblies have been theoretically established using molecular electrostatic potential (MEP) surface analysis and the combined QTAIM/NCI plot computational method.

## 2. Experimental Section

### 2.1. Materials and Methods

The chemicals used in the present study were purchased from commercial sources and used as received. Elemental analyses of the compounds were carried out using a Perkin Elmer 2400 Series II CHN/O analyzer. A Bruker ALPHA II Infrared spectrophotometer was used to record the FT-IR spectra of the compounds in the frequency range of 4000–500 cm$^{-1}$. The diffuse-reflectance electronic spectra of the compounds were recorded using a Shimadzu UV-2600 spectrophotometer. To record the solid state UV-Vis-NIR spectra of the compounds, $BaSO_4$ powder was used as a reference (100% reflectance). Room temperature magnetic moments were calculated at 300 K using a Sherwood Mark 1 Magnetic Susceptibility balance. Thermal analyses of the compounds were carried out with a Mettler Toledo TGA/DSC1 STAR$^e$ system under the flow of $N_2$ gas at a heating rate of 10 °C per min$^{-1}$.

### 2.2. Syntheses

2.2.1. Synthesis of [Cu(phen)$_2$Cl]NO$_3$·H$_2$O (Compound **1**)

$Cu(NO_3)_2$·3H$_2$O (0.241 g, 1 mmol) and $CaCl_2$·2H$_2$O (0.147 g, 1 mmol) were dissolved in 10 mL of methanol and mechanically stirred at room temperature for two hours. To the resulting solution, phen (0.396 g, 2 mmol) was added, and the solution was stirred for another hour (Scheme 1). The resulting solution was kept in cool conditions in a refrigerator for crystallization, from which blue block-shaped crystals were obtained after a few days. The yield was: 0.453 g (84%). The analytically calculated for the $C_{24}H_{18}ClCuN_5O_4$ were: C, 53.44%; H, 3.36%; and N, 12.98%. We found: C, 53.38%; H, 3.25%; and N, 12.90%. The FT-IR spectral data (KBr disc, cm$^{-1}$) were: 3452(s), 2825(w), 1588(s), 1427(s), 1382(s), 1374(s), 1351(m), 1145(w), 847(m), 718(m), and 580(w) (where: s, strong; m, medium; w, weak; br, broad; and sh, shoulder).

**Scheme 1.** Syntheses of compounds **1** and **2**.

2.2.2. Synthesis of [Cu$_2$(μ-Cl$_2$)Cl$_2$(Hdmpz)$_4$] (Compound **2**)

$Cu(NO_3)_2$·3H$_2$O (0.241 g, 1 mmol) and $CaCl_2$·2H$_2$O (0.147 g, 1 mmol) were mixed in deionized water (10 mL) and mechanically stirred for two hours. Hdmpz (0.184 g, 2 mmol) was then added to the resulting solution and stirred for another hour (Scheme 1). Then, the reaction mixture was kept undisturbed in cool conditions (below 4 °C) for crystallization, from which block-shaped blue single crystals were obtained after few days. The yield was: 0.548 g (84%). The analytically calculated for the $C_{20}H_{32}Cl_4Cu_2N_8$ were: C, 36.76%; H, 4.94%; and N, 17.15%. We found: C, 36.63%; H, 4.89%; and N, 17.10%. The FT-IR spectral data (KBr disc, cm$^{-1}$) were: 3345(s), 2924(w), 2841(w), 2726(w), 1573(m), 1466(w), 1267(m), 1160(w), 1045(s), 809(m), 702(w), 663(w), and 580(w).

### 2.3. Crystallographic Data Collection and Refinement

The crystallographic data collection of compounds **1** and **2** was carried out in a Bruker D8 Venture diffractometer with a Photon III 14 detector, using an Incoatec high brilliance IµS DIAMOND Cu tube equipped with Incoatec Helios MX multilayer optics. The Bruker APEX4 program was used for the data reduction and cell refinements [54]. SADABS was

used for the scaling and absorption corrections [54]. Using Olex2 [55], the structures were solved with the XT structure solution program [56] using intrinsic phasing and refined with the XL refinement package [56] using least squares minimization. All non-hydrogen atoms were refined with anisotropic thermal parameters by full-matrix least-squares calculations on $F^2$. Hydrogen atoms were inserted at calculated positions and refined as riders. The molecular structures and the packing diagrams were drawn using Diamond 3.2 [57]. The crystallographic results obtained for compounds **1** and **2** are summarized in Table 1.

**Table 1.** Crystallographic data and structure refinement details for compounds **1** and **2**.

| Crystal Parameters | 1 | 2 |
|---|---|---|
| Empirical formula | $C_{24}H_{18}ClCuN_5O_4$ | $C_{20}H_{32}Cl_4Cu_2N_8$ |
| Formula weight | 539.42 | 653.41 |
| Temperature (K) | 100.0 | 111.0 |
| Wavelength (Å) | 1.54178 | 1.54178 |
| Crystal system | Triclinic | Monoclinic |
| Space group | $P\bar{1}$ | $P2_1/c$ |
| $a/Å$ | 9.6570(9) | 8.6960(6) |
| $b/Å$ | 10.8732(10) | 13.4486(9) |
| $c/Å$ | 11.9997(11) | 11.8533(8) |
| $\alpha°$ | 68.329(2) | 90 |
| $\beta°$ | 70.758(2) | 106.117(2) |
| $\gamma°$ | 71.247(2) | 90 |
| Volume ($Å^3$) | 1076.48(17) | 1331.75(16) |
| Z | 2 | 2 |
| Calculated density ($g/cm^3$) | 1.664 | 1.629 |
| Absorption coefficient ($mm^{-1}$) | 2.965 | 5.872 |
| F (000) | 550.0 | 668.0 |
| Crystal size ($mm^3$) | $0.45 \times 0.21 \times 0.21$ | $0.22 \times 0.152 \times 0.108$ |
| $\theta$ range for data collection (°) | 8.166 to 136.698 | 10.59 to 133.154 |
| Index ranges | $-11 \le h \le 11,$ $-13 \le k \le 13,$ $-14 \le l \le 14$ | $-10 \le h \le 8,$ $-16 \le k \le 16,$ $-14 \le l \le 14$ |
| Reflections collected | 27,524 | 8827 |
| Unique data ($R_{int}$) | 3909 (0.0523) | 2336 (0.0359) |
| Refinement method | Full-matrix least-squares on $F^2$ | Full-matrix least-squares on $F^2$ |
| Data/restraints/parameters | 3909/0/320 | 2528/0/205 |
| Goodness-of-fit on $F^2$ | 1.162 | 1.188 |
| Final $R$ indices [$I > 2\sigma (I)$] $R1/wR2$ | 0.0412/0.1047 | 0.0435/0.1127 |
| $R$ indices (all data) $R1/wR2$ | 0.0413/0.1048 | 0.0439/0.1129 |
| Largest diff. peak and hole (e.$Å^{-3}$) | 0.65 and $-0.71$ | 1.35 and $-0.32$ |

CCDC163545 and 2163546 contain the supplementary crystallographic data for compounds **1** and **2**, respectively. These data can be obtained free of charge at http://www.ccdc. cam.ac.uk or from the Cambridge Crystallographic Data Centre (12Union Road, Cambridge CB2 1EZ, UK; fax: (+44) 1223-336-033; E-mail: deposit@ccdc.cam.ac.uk).

*2.4. Theoretical Methods*

The interaction energies of the compounds and supramolecular assemblies investigated in this work were computed at the RI-BP86-D3/def2-TZVP [58,59] level of theory using experimental geometries (only the H atoms were optimized) and the program Turbomole 7.2 [60]. We did not optimize the geometries because we were interested in estimating and characterizing the non-covalent forces as they stood in a solid state. Grimme's D3 dispersion [58] correction was used since it is convenient for the correct evaluation of non-covalent interactions, especially those involving π-systems, such as those investigated herein. The NCI plot [61] reduced density gradient (RGD) isosurfaces and QTAIM method [62] were combined in the same representation to characterize the non-covalent

interactions by means of the MULTIWFN program [63], and these were represented using VMD software [64].

## 3. Results

### 3.1. Synthesis and General Aspects

Compound **1** was prepared by the reaction between $Cu(NO_3)_2 \cdot 3H_2O$, $CaCl_2$ and phen (1:1:2) at room temperature in methanol medium. Similarly, compound **2** was synthesized by the reaction between $Cu(NO_3)_2 \cdot 3H_2O$, $CaCl_2$ and Hdmpz (1:1:2) at room temperature in de-ionized a water medium. Both the compounds are soluble in water and in common organic solvents. The compounds exhibited room temperature magnetic moment values of 1.76 and 1.74 BM, respectively, indicating the presence of one unpaired electron per Cu(II) center [65]. Crystallographic data collection at a lower temperature can improve the quality of the data sets, thereby resulting in more accurately solved crystal structures [66]. Simple synthetic procedures in aqueous media were followed for compounds **1** and **2**, and the crystal structures were redetermined at low temperatures and with different unit cell parameters (Tables S1 and S2) than the previously reported compounds [67–71]. Improved R and wR values for the compounds implied that there was improved crystallographic data collection for the compounds [72,73]. We also explored non-covalent $(\pi–\pi)_1/(\pi–\pi)_2/(\pi–\pi)_1$ and anion–$\pi$/($\pi–\pi$)/anion–$\pi$ ternary assemblies in compound **1** and unconventional type I Cl···Cl interactions in compound **2** (vide infra). The supramolecular assemblies of the compounds were further established using various computational tools.

### 3.2. Crystal Structure Analysis

Figure 1 depicts the molecular structure of compound **1**. Selected bond lengths and bond angles are tabulated in Table 2. Compound **1** crystallized in the triclinic $P\overline{1}$ space group. The Cu(II) center in compound **1** was penta-coordinated with two bidentate phen moieties and one chloride moiety. The asymmetric unit of compound **1** contained one nitrate and one lattice water molecule. The coordination geometry around the Cu(II) center was square pyramidal, as evidenced by the trigonality index ($\tau$) value of 0.25 [74,75]. The Cl atom (Cl1) occupied the apical position, whereas the equatorial positions were occupied by the four N atoms (N1, N1′, N10, and N10′) from the two phen moieties. The Cu–Cl and average Cu–N bond lengths were comparable to those of the already reported Cu(II) compounds [76,77].

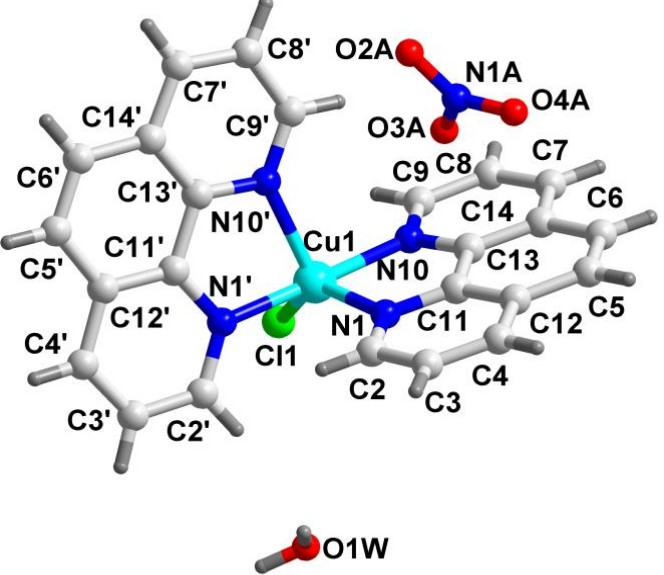

**Figure 1.** Molecular structure of [Cu(phen)$_2$Cl]NO$_3$·H$_2$O (**1**).

**Table 2.** Selected bond lengths (Å) and bond angles (°) of the Cu(II) centers in compounds **1** and **2**.

| Compound **1** | | | |
|---|---|---|---|
| Cu1–Cl1 | 2.2937(6) | N10–Cu1–N1 | 81.63(8) |
| Cu1–N1 | 2.0827(2) | N10–Cu1–N1#1 | 175.54(8) |
| Cu1–N10 | 1.987(2) | N10–Cu1–N10#1 | 96.83(8) |
| Cu1–N1#1 | 1.988(2) | N1#1–Cu1–Cl1 | 91.54(6) |
| Cu1–N10#1 | 2.137(2) | N1#1–Cu1–N1 | 95.25(8) |
| N1–Cu1–Cl1 | 139.47(6) | N1#1–Cu1–N10#1 | 80.79(8) |
| N1–Cu1–N10#1 | 104.12(7) | N10#1–Cu1–Cl1 | 116.41(5) |
| N10–Cu1–Cl1 | 92.90(6) | | |
| Compound **2** | | | |
| Cu1–Cl1#1 | 2.6558(1) | N1–Cu1–Cl1#1 | 100.44(9) |
| Cu1–Cl | 2.3229(1) | Cl1–Cu1–Cl1#1 | 84.90(3) |
| Cu1–Cl2 | 2.2983(1) | N1–Cu1–Cl2 | 89.12(9) |
| Cu1–N1 | 2.013(3) | N1#–Cu1–Cl1#1 | 99.07(9) |
| Cu1–N1# | 2.003(3) | N1#–Cu1–Cl1 | 89.41(9) |
| Cl2–Cu1–Cl1#1 | 98.65(3) | N1#–Cu1–Cl2 | 162.25(9) |
| Cl2–Cu1–Cl1 | 91.27(3) | N1#–Cu1–N1 | 88.56(1) |
| N1–Cu1–Cl1 | 174.53(9) | O5–Mn2–O5 | 86.70(6) |

#1 2-X, 1-Y, 1-Z.

The lattice nitrate moieties and the complex cationic moieties of compound **1** were involved in the formation of the anion–π/π–π/π–anion assembly along the crystallographic ac plane (Figure 2a).The O3A atom of the lattice nitrate was involved in the anion−π interactions with the pyridyl ring of the coordinated phen, having a O3A···Cg distance of 3.56 Å, while the corresponding angle between the O3A−Cgplane of the ring was found to be 98.4° [78–80]. Aromatic π-stacking interactions were observed between the aromatic rings of the coordinated phen, having centroid–centroid separations of 3.53 and 3.87 Å, respectively. A C–H···Cl hydrogen-bonding interaction was observed between the coordinated Cl atom (Cl1) and the –CH moiety of the phen, having a C5–H5···Cl1 distance of 2.92 Å. The O3A atom of the lattice nitrate anion was also involved in the C–H···O hydrogen-bonding interactions, having a C9–H9···O3A distance of 2.80 Å. This anion–π/π–π/π–anion assembly observed in the crystal structure of compound **1** was further studied theoretically (vide infra). These anion–π/π–π/π–anion assemblies, along with lattice water molecule of compound **1**, were involved in the formation of the layered assembly along the crystallographic ac plane, which was stabilized by the C–H···O, C–H···Cl, O–H···Cl, and O–H···O hydrogen-bonding and the non-covalent C–H···π, π-π interactions (Figure 2b). The lattice nitrate and water molecule were interconnected via the O–H···O hydrogen-bonding interactions, having a O1W–H1WA···O2A distance of 1.94 Å. Similarly, the O4A atom of the lattice nitrate was involved in the C–H···O hydrogen-bonding interaction, having a C4–H4···O4A distance of 2.44 Å. The C–H···Cl hydrogen-bonding interactions were observed between the coordinated Cl atom (Cl1) and the –CH moiety of the phen, having a C6–H6···Cl1 distance of 2.98 Å. An O–H···Cl hydrogen-bonding interaction was observed between the coordinated Cl1 atom and the O1W lattice water molecule, having a O1W–H1WB···Cl1 distance of 2.33 Å. The –C3H3 moiety of the phen was involved in the C–H···π interaction, having C3···Cg and H3···Cg separation distances of 3.83 and 3.12 Å, respectively (Cg was the ring centroid defined by the atoms C5, C6, and C11–C14). Aromatic π-stacking interactions were observed between the phenyl moieties of the coordinated phen, having centroid–centroid separations of 3.62 and 3.61 Å, respectively.

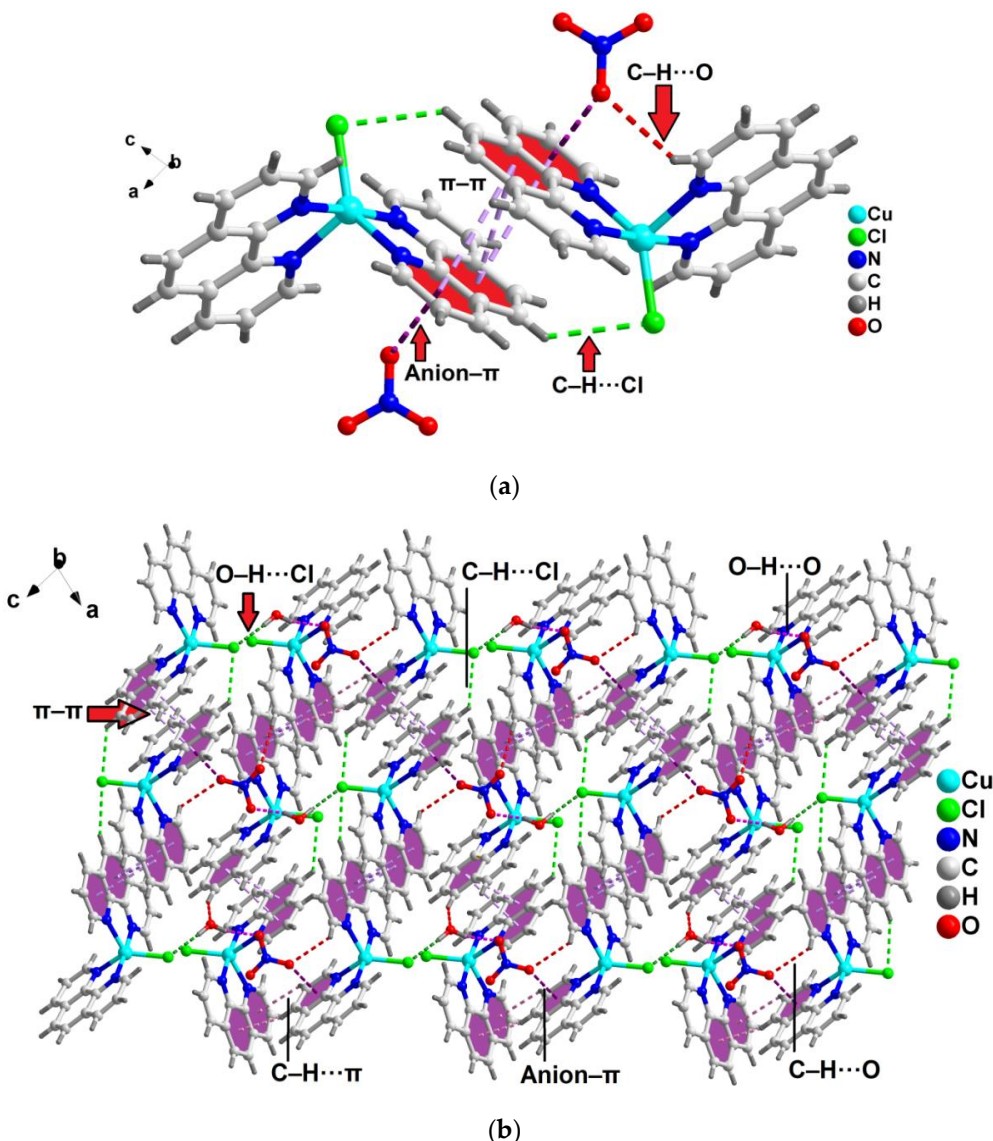

**Figure 2.** (**a**) Formation of the anion–π/π–π/π–anion assemblies in compound **1** involving the uncoordinated nitrate anions. (**b**) Layered assembly of compound **1** along the crystallographic ac plane aided by the anion–π/π–π/π-anion assemblies.

As shown in Figure S1, a 1D chain of compound **1** was formed via the C–H⋯Cl and π-stacking interactions. Cl1 and the –CH moieties of the coordinated phen were involved in the C–H⋯Cl hydrogen-bonding interactions, having C6′–H6′⋯Cl1 and C5′–H5′⋯Cl1 distances of 2.98 and 2.92 Å, respectively. Moreover, aromatic π-stacking interactions were observed between the neighboring phen moieties, having centroid–centroid separations of 3.61, 3.62, 3.87, 3.96, 3.46 and 3.53 Å, respectively.

The layered architecture of compound **1** was stabilized by the non-covalent C–H⋯π interactions (Figure S2). C–H⋯π interactions were observed between the –C4H4 moiety of the phen with the π-electron of the neighboring phen. The distance from the carbon atom (C4) to the π-electron of the phen was found to be 3.45 Å, whereas the distance from the hydrogen atom (H4) to the π-electron of the phen was found to be 2.53 Å. Moreover, the corresponding C–H⋯π angle was 120.4°, which indicated the significance of the interaction [81].

Figure 3 represents the molecular structure of compound **2**. Selected bond lengths and bond angles are tabulated in Table 2. Compound **2** crystallized in the monoclinic P2$_1$/c space group. Compound **2** was a dimeric Cu(II) compound-bridged by two Cl ions.

The coordination geometry around the Cu(II) centers were penta-coordinated with two pyrazole nitrogen atoms (N1 and N1′) of two Hdmpz moieties, two bridging Cl ions (Cl1) and one monodentate Cl atom (Cl2). Compound 2 possessed a crystallographic inversion center of symmetry which was present at the center of the compound. The coordination geometries around the Cu(II) centers in compound **2** were distorted square pyramidal, as evidenced by the value of trigonality index (0.12), which was consistent with the square pyramidal geometry. The Cl atom (Cl2) was present at the axial site, whereas the equatorial sites were occupied by two nitrogen atoms (N1, N1′) and two bridging Cl atoms (Cl1). The average Cu–Cl and Cu–N bond lengths were in good agreement with the previously reported similar compounds [82].

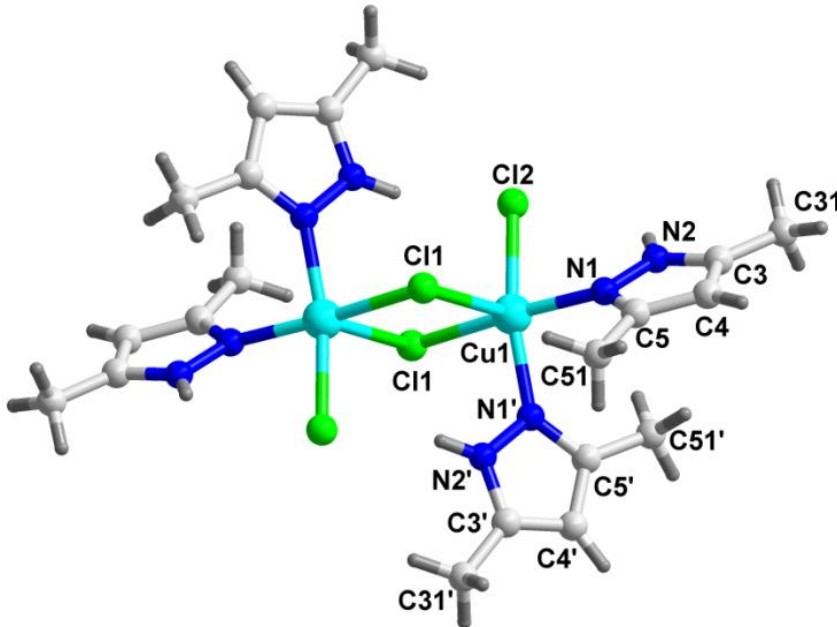

**Figure 3.** Molecular structure of $[Cu_2(\mu-Cl_2)Cl_2(Hdmpz)_4]$ (compound **2**).

As shown in Figure 4, the adjacent dinuclear units of compound **2** were interconnected via the N–H···Cl and C–H···Cl hydrogen-bonding, and non-covalent C–H···C and unconventional Cl···Cl contacts formed the 1D supramolecular chain along the crystallographic a-axis. N–H···Cl contacts were observed, involving the coordinated Cl atom (Cl2) and the –NH (N2H2) moiety of the Hdmpz, having N–H···Cl bond distances of 2.68 and 2.32 Å, respectively. Similarly, N–H···Cl and C–H···Cl interactions were also observed, having N2–H2···Cl1 and C31–H31C···Cl2 distances of 2.99 and 3.02 Å, respectively. Non-covalent C–H···C interactions [83] were involved in the stabilization of the 1D chain of compound **2** (Table S3). Moreover, unconventional Cl···Cl interactions were also observed between the Cl atoms of the neighboring monomeric units, having Cl···Cl separation distances of 3.58 Å. The Cl···Cl interactions observed in the 1D supramolecular chain of compound **2** could be considered type I, with the corresponding angles θ1 = θ2 = 91.5°. The type I Cl···Cl interactions were explored in a Mn(II) coordination polymer, namely, $[Mn(tcpa)_2(bipy)]_n$ (where Htcpa and bipy represent 3,5,6-trichloropyridine-2-oxyacetic acid and 2,2′-bipyridine, respectively) [84]. The unconventional Cl···Cl interactions (type I) were further established using the NCI plot computational tool (vide infra).

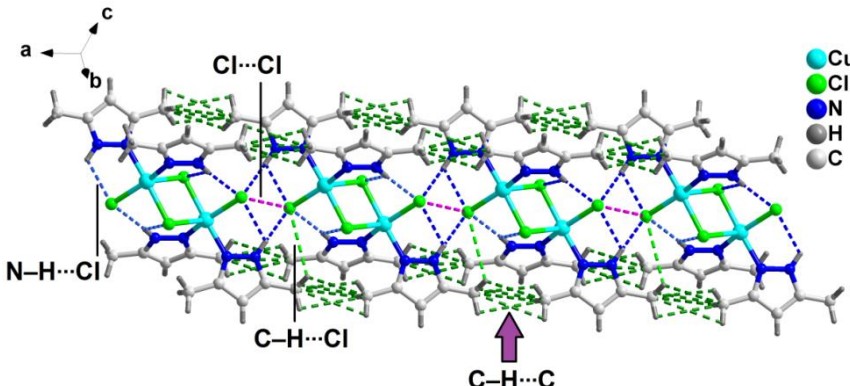

**Figure 4.** The 1D supramolecular chain of compound **2** assisted via the N–H···Cl, C–H···Cl hydrogen-bonding and the non-covalent C–H···C and unconventional Cl···Cl interactions.

Figure S3 represents the layered assembly of compound **2** along the crystallographic ac plane, as assisted by the non-covalent C–H···C interactions. The methyl groups of the Hdmpz moieties of the neighboring monomeric units were involved in the non-covalent C–H···C interactions (Table S3).

The neighboring monomeric units of compound **2** were also interconnected via the intermolecular C–H···Cl hydrogen-bonding and non-covalent C–H···C interactions to form a layered assembly along the crystallographic ab plane (Figure 5). The coordinated Cl (Cl2) atom was involved in the C–H···Cl hydrogen-bonding interactions with the –CH moiety of the Hdmpz, having a C51–H51B···Cl2 distance of 2.90 Å (Table 3). Moreover, the Cl1 atom was also involved in the C–H···Cl hydrogen-bonding interactions with the –CH moiety of the Hdmpz, having C31–H31E···Cl1 and C31–H31B···Cl1 distances of 2.99 and 2.89 Å, respectively. Furthermore, non-covalent C–H···C contacts were also observed in the layered assembly of compound **2** (Table S3).

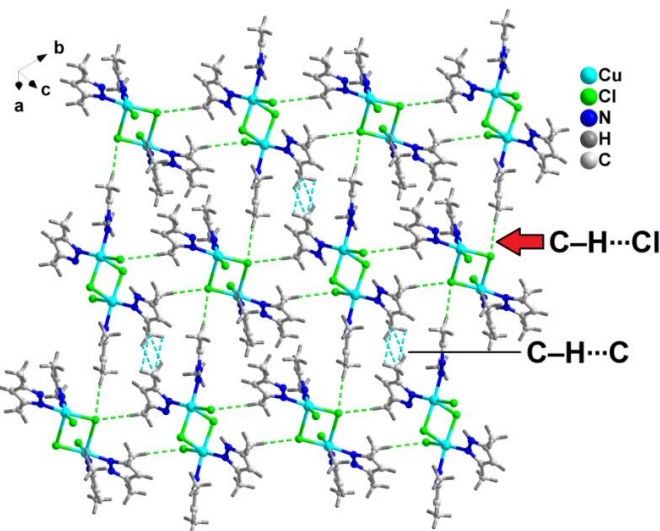

**Figure 5.** Layered assembly of compound **2** along the crystallographic ab plane, as assisted by the C–H···Cl hydrogen-bonding and non-covalent C–H···C interactions.

**Table 3.** Selected hydrogen bond distances (Å) and angles (°) for compounds **1** and **2**.

| D–H···A | d(D–H) | d(D···A) | d(H···A) | <(DHA) |
|---|---|---|---|---|
| **1** | | | | |
| C6′–H6′···Cl1 | 0.94 | 3.873(2) | 2.98 | 156.7 |
| C5′–H5′···Cl1 | 0.94 | 3.649(2) | 2.92 | 133.7 |
| C9–H9···O3A | 0.94 | 3.215(2) | 2.80 | 107.4 |
| C4–H4···O4A | 0.94 | 3.345(2) | 2.44 | 157.5 |
| O1W–H1WA···O2A | 0.87 | 2.792(2) | 1.94 | 164.3 |
| O1W–H1WB···Cl1 | 0.87 | 3.203(3) | 2.33 | 174.4 |
| **2** | | | | |
| N2–H2···Cl2 | 0.87 | 3.114(2) | 2.68 | 111.3 |
| N2–H2···Cl2 | 0.87 | 3.180(2) | 2.32 | 164.9 |
| N2–H2···Cl1 | 0.87 | 3.517(2) | 3.00 | 119.4 |
| C51–H51B···Cl1 | 0.97 | 3.820(2) | 2.89 | 157.2 |
| C31–H31E···Cl1 | 0.97 | 3.895(2) | 3.00 | 151.8 |
| C31–H31B···Cl1 | 0.97 | 3.859(2) | 2.88 | 171.3 |

*3.3. Spectral Studies*

3.3.1. FT-IR Spectroscopy

The FT-IR spectral analyses of the compounds were completed in detail (see ESI; Figure S4). The absorption bands due to coordinated phen and Hdmpz moieties were obtained at the expected positions [85–89]. The presence of uncoordinated water and nitrate anions in the compounds could also be corroborated by their respective peaks [90,91].

3.3.2. Electronic Spectroscopy

Figures S5 and S6 depict the solid and aqueous phase electronic spectra of compounds 1 and 2, respectively. The electronic spectra of the compounds suggested the presence of Cu(II) centers in the compounds [92]. The similar absorption bands in both the phases indicated that the compounds did not undergo any distortion in the aqueous phase [93–95].

*3.4. Thermogravimetric Analysis*

We recorded the thermogravimetric curves of compounds **1** and **2** in the temperature range of 25–800 °C under an $N_2$ atmosphere at a heating rate of 10 °C/min (Figure S7). For compound **1**, in the temperature range of 50–160 °C, the compound underwent a loss of water molecules of crystallization (obs. = 3.92%, calcd. = 3.39%) [96,97]. In the temperature range of 161–375 °C, the lattice nitrate and one phen moiety underwent decomposition, with a weight loss of 44.53% (calcd. = 44.52%) [98–100]. In the 376–790 °C range, Cl and half of the phen moiety underwent decomposition, with a 25.34% weight loss (calcd. = 23.68%) [97,99,100]. Beyond the temperature, the compounds decomposed in an unidentified manner. Compound **2** decomposed in a single step in the range of 51–370 °C, with the loss of four Cl and four Hdmpz moieties (obs. = 78.24%, calcd. = 79.34%) [96,101]. Compound **2** was likely to degrade to CuO at temperatures above 500 °C (Figure S8) [102].

*3.5. Theoretical Study*

The theoretical study was focused on the non-covalent interactions present in the compounds, particularly the anion-π/π-π/π-anion assemblies in compound **1** and the H bonds and Cl···Cl interactions in compound **2**. First, we computed the MEP surfaces of compounds **1** and **2** to investigate the most electrophilic and nucleophilic parts of the molecules. The MEP surfaces are represented in Figure 6, which discloses that the MEP minimum was located at the nitrate anion, as expected (−59 kcal/mol), followed by the chlorido ligand (−37 kcal/mol). The MEP maximum was located at the aromatic H atoms of the coordinated phen (25 kcal/mol); therefore, the formation of the H bonds between the aromatic protons and the counter ions (nitrate and chloride) were the most electrostatically favored contacts. Interestingly, the MEP values were small and positive

over the aromatic rings of the phen ligand (ranging from 3 to 6 kcal/mol). Such small MEP values facilitated the formation of the π-stacking interactions since the electrostatic repulsion was small and compensated by other attractive forces such as dispersion and polarization. It is interesting to highlight that the MEP value over the five-membered chelate ring was negative ($-5$ kcal/mol), thus favoring the formation of the antiparallel π-stacking interactions since the negative chelate ring was approximate to the positive central ring of the phen, as observed in compound **1**. Moreover, the antiparallel arrangement of the dipoles significantly increased the binding energy, as we previously demonstrated [103]. The MEP of compound **2** showed that the MEP maximum was located at one of the NH groups (37 kcal/mol) (the one that did not forming an intramolecular N–H···Cl bond). The MEP was also large and positive at the H atoms of the $CH_3$ groups (~18 kcal/mol). The MEP minimum was located at the coordinated chlorido ligands ($-48$ kcal/mol); therefore, the most-favored interaction corresponded to the N–H···Cl bonds, as observed in the solid state of compound **2**.

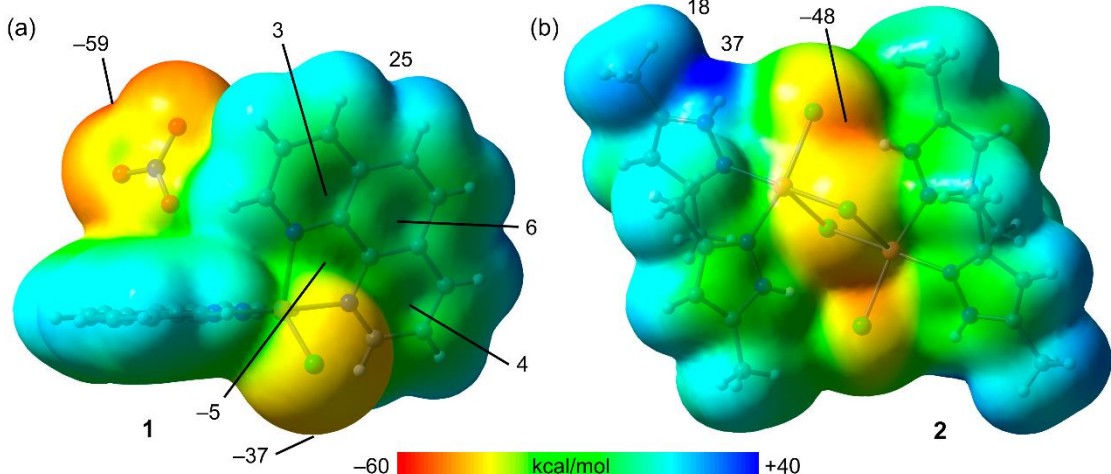

**Figure 6.** MEP surfaces of compounds **1** (**a**) and **2** (**b**) at the RI-BP86-D3/def2-TZVP level of theory. The isovalue was 0.001 a.u., and the energies at selected points are indicated in kcal/mol.

Figure 7 shows the anion–π/π–π/anion–π assembly observed in compound **1**, where two nitrate anions established an anion–π interaction with a π-stacked dimer, with a large overlap of the π-systems of the phen rings. The anion–π interaction likely reinforced the π-stacking by reducing the electrostatic repulsion between the π-acidic phen rings, i.e., the phen rings became electron-deficient upon coordination with the Cu(II) metal center. Such an effect was compensated by the anion–π interaction at the opposite side of the ring, as previously demonstrated in the literature [104,105]. A model of the anion–π/π–π/anion–π assembly is given in along with the interaction energy (computed as the π-stacked dimer) of $\Delta E_1 = -23.6$ kcal/mol, confirming that the antiparallel π-stacking was very strong. The assembly was further strengthened by the formation of the C–H···Cl interactions, which was in good agreement with the MEP surface analysis discussed above. In order to estimate the contribution of the C–H···Cl bonds, we computed a mutated model where the chlorido ligands were replaced by hydrido ligands. Consequently, the H bonds were not formed, and the interaction energy of the mutated dimer was reduced to $\Delta E_2 = -19.1$ kcal/mol, which corresponded to the π-stacking interaction, and the difference (–4.5 kcal/mol) corresponded to the C–H···Cl hydrogen bonding. Therefore, the π-stacking interaction was the dominant force in this assembly.

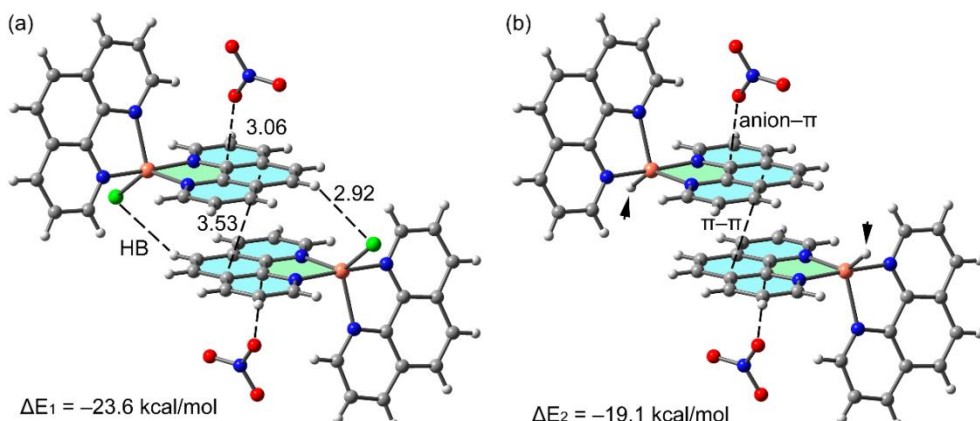

**Figure 7.** Complete (**a**) and mutated (**b**) theoretical models used to evaluate the π-stacking and C–H···Cl H bonds in compound **1**. The distances were measured in Å.

An H-bonded dimer extracted from the 1D supramolecular assembly of compound **2** was analyzed theoretically, as shown in Figure 8. We used red dashed lines to represent the intra-molecular N–H···Cl bonds and black dashed lines to represent the intermolecular H bonds. Two different types of H bonds observed (N–H···Cl and C–H···Cl) were in good agreement with the MEP analysis shown in Figure 6b. The dimerization energy was quite large and negative ($\Delta E_3 = -24.7$ kcal/mol), thus evidencing the strong nature of these H bonds. In order to estimate the energy of the N–H···Cl H-bonds, we used a mutated dimer, where one of the methyl groups of the organic ligand was replaced by H atoms (see the small arrows in Figure 8b).

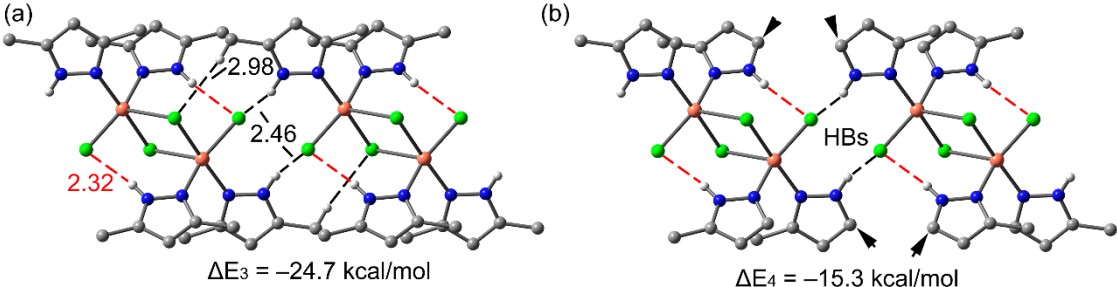

**Figure 8.** Complete (**a**) and mutated (**b**) theoretical models used to evaluate the N–H···Cl and C–H···Cl H bonds in compound **2**. The distances were measured in Å. The H atoms have been omitted for clarity, apart from those that formed H bonds.

The interaction energy of the mutated dimer was reduced to $\Delta E_4 = -15.3$ kcal/mol, corresponding to the strength of the N–H···Cl bonds, and the difference (9.4 kcal/mol) could be attributed to the C–H···Cl and the additional van der Waals interactions, as discussed below.

Finally, a combination of the QTAIM and NCI plot (RDG isosurfaces) methods was used to characterize the interactions in the assemblies of compounds **1** and **2**. The anion–π interaction was characterized by a bond critical point (CP, represented by a red sphere) and a bond path (represented by orange lines) connecting one O atom of nitrate to one C atom of the coordinated phen. The anion was further connected to the adjacent phen ligand by a C–H···O H bond. The π-stacking interaction was characterized by six bond CPs and bond paths interconnecting several atoms of phen. Finally, the QTAIM also confirmed the existence of the C–H···Cl interactions, characterized by a bond CP and a bond path interconnecting the H and Cl atoms.

All these interactions were also revealed by the RDG isosurfaces. It is interesting to highlight that the NCI plot RDG isosurface that characterized the antiparallel π-stacking

interactions involved the phen moieties (vide supra). For compound **2**, the distribution of the critical points showed that each H bond was characterized by a corresponding bond CP, bond path and green isosurface. Remarkably, the NCI plot analysis also evidenced a green isosurface between the Cl atoms, thus confirming the existence of a Cl···Cl contact and suggesting its weak attractive nature. Moreover, there were also several green RDG isosurfaces located between the methyl groups and the ligands, disclosing the existence of van der Waals interactions and explaining the large reduction in the interaction energy of compound **2** when the mutated dimer was used (see Figure 9). The density, Laplacian of density and energy densities at the bond CPs are given in Table 4. It could be observed that in all cases, the values of ρ were less than 0.01 a.u., and Gr > |Vr|, as is typical in closed shell non-covalent interactions.

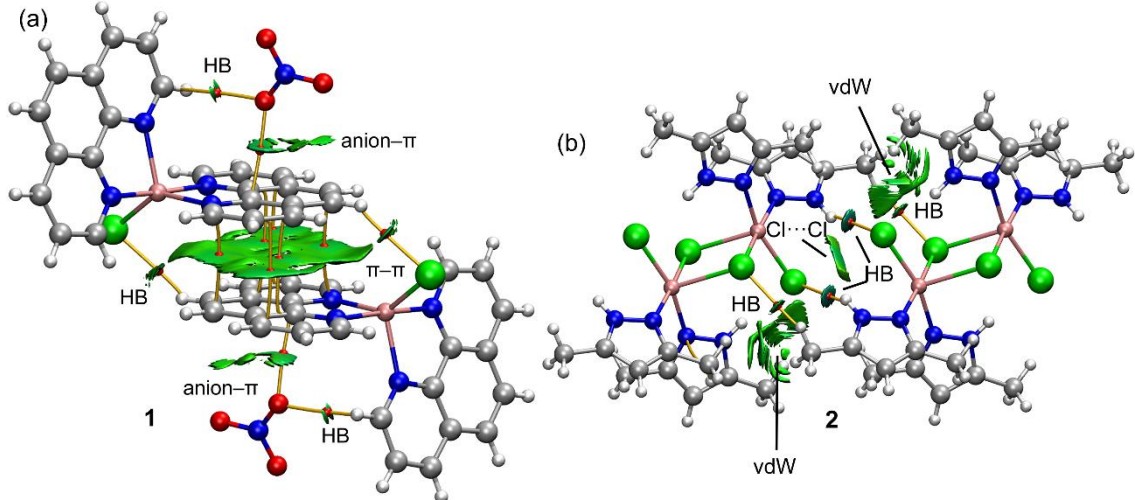

**Figure 9.** Combined QTAIM (bond CPs shown in red and bond paths shown as orange lines) and RDG isosurface analyses of the assemblies of compounds **1** (**a**) and **2** (**b**).

**Table 4.** QTAIM parameters at the bond CPs depicted in Figure 9. The density (ρ), Lagrangian kinetic energy density (Gr), potential energy density (Vr), total energy density (Hr) and Laplacian ($\nabla^2\rho$) are given in a.u.

| Interaction | ρ | Gr | Vr | Hr | $\nabla^2\rho$ |
|---|---|---|---|---|---|
| Dimer compound **1** (Figure 9a) | | | | | |
| CH···Cl | 0.00695 | 0.00432 | −0.00324 | 0.00108 | 0.02160 |
| π···π (N···C) | 0.00526 | 0.00335 | −0.00238 | 0.00098 | 0.01732 |
| π···π (C···C) | 0.00625 | 0.00359 | −0.00274 | 0.00085 | 0.01772 |
| π···π (C···C) | 0.00541 | 0.00321 | −0.00232 | 0.00089 | 0.01638 |
| anion···π | 0.00852 | 0.00634 | −0.00477 | 0.00157 | 0.03162 |
| CH···O (nitrate) | 0.00615 | 0.00431 | −0.00305 | 0.00125 | 0.02222 |
| Dimer compound **2** (Figure 9b) | | | | | |
| CH···Cl | 0.00593 | 0.00352 | −0.00264 | 0.00089 | 0.01764 |
| NH···Cl | 0.01441 | 0.01014 | −0.00768 | 0.00246 | 0.05041 |

## 4. Conclusions

Two coordination compounds of Cu(II) involving N-donor ligands (phen and Hdmpz) were synthesized and characterized using elemental analysis, TGA, spectroscopic (FT-IR and electronic) techniques and single crystal XRD. The lattice nitrate anion and the coordinated phen moieties of compound **1** were involved in unusual anion–π/(π–π)/anion–π assemblies which stabilized its crystal structure. The crystal structure analysis of compound **2** revealed the presence of unusual Cl···Cl contacts involving the coordinated Cl

moieties. Moreover, DFT calculations were performed to evaluate the strengths of the antiparallel π-stacking interactions, along with anion-π contacts, in compound **1**, evidencing the role of cooperative anion–π/(π–π)/anion–π assemblies towards the stabilization of the compound. Similarly, in compound **2**, the H-bonding contacts were found to be the structure-guiding non-covalent forces, thereby stabilizing its crystal structure. These interactions were further characterized using the MEP surface, NCI plot and QTAIM computational tools.

**Supplementary Materials:** The following supporting information can be downloaded at: https://www.mdpi.com/article/10.3390/cryst13030517/s1, Figure S1: 1D supramolecular chain of compound **1** assisted by C–H⋯Cl and π-stacking interactions; Figure S2: Layered assembly of compound **1** along the crystallographic ac plane; Figure S3: Layered assembly of compound **2** assisted by non-covalent C–H⋯C interactions; Figure S4: FTIR spectra of compounds **1** and **2**; Figure S5: (a) UV-Vis-NIR spectrum of **2** (b) UV-Vis spectrum of **2** in water; Figure S6: (a) UV-Vis-NIR spectrum of **3** (b) UV-Vis spectrum of **3**; Figure S7: TGA curves of compounds **1** and **2**; Figure S8: Decomposition of various ligands in various steps of compounds **1–2** in TGA analysis, Table S1: Comparison of crystal parameters of compound **1** with the already reported compounds; Table S2: Comparison of crystal parameters of compound **2** with the already reported compound; Table S3: Selected parameters for C–H⋯C interactions in compound **2**. References [92–95] are cited in the supplementary materials.

**Author Contributions:** Conceptualization, A.F. and M.K.B.; methodology, A.F. and M.K.B.; software, A.F. and R.M.G.; formal analysis, A.F.; investigation, P.S. and R.M.G.; data curation, M.B.-O.; writing—original draft preparation, P.S. and M.K.B.; writing—review and editing, M.K.B.; visualization, A.F.; supervision, M.K.B.; project administration, A.F. and M.K.B.; funding acquisition, A.F. and M.K.B. All authors have read and agreed to the published version of the manuscript.

**Funding:** Financial support was provided by ASTEC, DST, Govt. of Assam (grant number ASTEC/S&T/192(177)/2020-2021/43) and the Gobierno de Espana, MICIU/AEI (project number PID2020-115637GB-I00), all of whom are gratefully acknowledged. The authors thank IIT-Guwahati for the TG data.

**Data Availability Statement:** Not applicable.

**Conflicts of Interest:** The authors declare no conflict of interest. The funders had no role in the design of the study; in the collection, analyses, or interpretation of data; in the writing of the manuscript; or in the decision to publish the results.

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
