# Peer review of "Cooperative Ternary Assemblies Involving Anion–π/π–π/Anion–π Assemblies and Unconventional Cl⋯Cl Interactions in Cu(II) Coordination Compounds: Experimental and Theoretical Studies"

_crystals, doi:10.3390/cryst13030517_

Round 1

Reviewer 1 Report

This work is an interesting and detail study of the synthesis ad structures of three Phen copper(II) complexes. The work can be published after correction of some points.

1. Line 51

"miryad attempts..." and we see only two references to ane authors' team. This is really strange. Please, add more fresh references to different authors to verify your statement.

2. In this paragraph (lines 47-70), dihydrogen bonds and other non-covalent interactions (Cl...Cl are the closest to the present study) should be mentioned.

For example,

https://doi.org/10.1021/acs.chemrev.6b00091

doi:10.3390/cryst9070330

https://doi.org/10.1134/S0036023620040105

https://doi.org/10.1002/ejic.201000157

https://doi.org/10.1351/PAC-REC-12-05-10

3. Line 74

Again, the statement is very general "Phen is capable of forming stable coordination complexes...", but references are too scarcely. Please add more references, for example

10.1134/S0036023620040026

https://doi.org/10.1134/S0036023622100278

https://doi.org/10.1016/j.ccr.2010.04.008

https://doi.org/10.3390/inorganics10070099

4. Line 115

In the synthesis of compounds 1 and 2, the authors used copper(II) nitrate and calcium chloride as a source of chloride anions. We advise the authors to see the reference https://doi.org/10.1134/S0036023620090119, where it was indicated that when the authors performed their copper complexation with Bipy in dichloroethane, CuBipy2Cl+ cation was formed; thus, the C2H4Cl2 solvent was the source of Cl- anions. I think the authors could use this approach to introduce Cl- anions in their future papers (if no water system could be appropriate).

5. Line 190

Did the authors performe their reactions with another Cu : Phen ratio? If the Cu : Phen ratio would be 1 : 3, can the tris-chelate [Cu(Phen)3]2+ complex form? With Cl- or NO3- as a counterion? I think this is interesting fact for the mechanism of the complexation process. 

6. I want to see more chemistry in Scheme 1. Cu(NO3)3 should dissociate in the first stage to Cu2+ and [NO3]-; then the authors add Cl- ions. It seems that [Cu(H2O)6]2+ aquacomplex should form. After addition of Phen, some H2O molecules seem to be replaced by Phen and Cl- forming neutral complex. Therefore, the information about different ratio of the initial components would be interesting and allow to discuss the mechanism.

7. Lines 200-201

As you indicated that crystal 3 was already known, is your synthesis of complex 3 the same as those mentioned in [52]?

8. Lines 351-355, 427-436 should be shortened. May be they can be present in Supplementary and Tables, and in the text they should not be discussed in so details. 

Author Response

First, we would like to thank this reviewer for his/her careful reading of the manuscript, corrections and suggestions. The changes made are detailed below:

This work is an interesting and detail study of the synthesis ad structures of three Phen copper(II) complexes. The work can be published after correction of some points.

  1. Line 51

"miryad attempts..." and we see only two references to ane authors' team. This is really strange. Please, add more fresh references to different authors to verify your statement.

Reply: We have now incorporated a few fresh references in the revised manuscript.

  1. In this paragraph (lines 47-70), dihydrogen bonds and other non-covalent interactions (Cl...Cl are the closest to the present study) should be mentioned.

For example,

https://doi.org/10.1021/acs.chemrev.6b00091

doi:10.3390/cryst9070330

https://doi.org/10.1134/S0036023620040105

https://doi.org/10.1002/ejic.201000157

https://doi.org/10.1351/PAC-REC-12-05-10

Reply: We have mentioned about dihydrogen bonds and other non-covalent interactions in the introduction section of the revised manuscript and cited the references as suggested by the esteemed reviewer.

  1. Line 74

Again, the statement is very general "Phen is capable of forming stable coordination complexes...", but references are too scarcely. Please add more references, for example

10.1134/S0036023620040026

https://doi.org/10.1134/S0036023622100278

https://doi.org/10.1016/j.ccr.2010.04.008

https://doi.org/10.3390/inorganics10070099

Reply: We have incorporated more appropriate references for the statement as suggested.

  1. Line 115

In the synthesis of compounds 1 and 2, the authors used copper(II) nitrate and calcium chloride as a source of chloride anions. We advise the authors to see the reference https://doi.org/10.1134/S0036023620090119, where it was indicated that when the authors performed their copper complexation with Bipy in dichloroethane, CuBipy2Cl+ cation was formed; thus, the C2H4Cl2 solvent was the source of Cl- anions. I think the authors could use this approach to introduce Cl- anions in their future papers (if no water system could be appropriate).

Reply: We thank the esteemed reviewer for the valuable suggestion and we will do that for crystallization of compounds for future manuscripts.

  1. Line 190

Did the authors perform their reactions with another Cu : Phen ratio? If the Cu : Phen ratio would be 1 : 3, can the tris-chelate [Cu(Phen)3]2+ complex form? With Cl- or NO3- as a counterion? I think this is interesting fact for the mechanism of the complexation process.

Reply: We agree with the esteemed reviewer. We already performed the reactions using different ratios; but unfortunately we could not crystallize the compounds.

  1. I want to see more chemistry in Scheme 1. Cu(NO3)3 should dissociate in the first stage to Cu2+ and [NO3]-; then the authors add Cl- ions. It seems that [Cu(H2O)6]2+ aquacomplex should form. After addition of Phen, some H2O molecules seem to be replaced by Phen and Cl- forming neutral complex. Therefore, the information about different ratio of the initial components would be interesting and allow to discuss the mechanism.

Reply: As already mentioned, we have already tried, but unfortunately we are not able to crystallize the compounds.

  1. Lines 200-201

As you indicated that crystal was already known, is your synthesis of complex 3 the same as those mentioned in [52]?

Reply: The synthetic methods are different and this has been mentioned in the revised manuscript.

  1. Lines 351-355, 427-436 should be shortened. May be they can be present in Supplementary and Tables, and in the text they should not be discussed in so details.

Reply: We have shortened the text accordingly. The parameters for C‒H∙∙∙C interactions have been tabulated in ESI. 

Reviewer 2 Report

The manuscript by Antonio Frontera et al describes synthesis and characterization of three copper(II) coordination compounds, two with phenanthroline and one with Hdmpz ligand. The characterization of compounds is complete and the study in general performed in a thorough manner. Nevertheless, the authors should resolve the following issues prior to publication.

i) The authors should comment upon the choice of ligands, i.e., phen vs. Hdmpz. Without some kind of explanation, a common point (or a difference) between the phen and the Hdmpz complexes is not obvious.

ii) Compounds 1 and 2 differ in the content of the water molecules of crystallization, apart from that, the coordination species are, at least as far as the content is concerned, the same. When describing the crystal structures of 1 and 2, the authors should focus also on the similarities or differences between the complex species [Cu(phen)2Cl]+ of 1 and 2. Perhaps a drawing with overlaid [Cu(phen)2Cl]+ ions of 1 and 2 should be included.

iii) The description of the supramolecular structures in 13 is too detailed and too lengthy and as such very difficult to follow. I suggest to have it shortened.

iv) lines 521 and 526

When describing the TG curves of 1 and 2, the authors stated that water molecules were decomposed. Such a formulation is wrong. The compounds underwent the loss of water molecules of crystallization. Please correct.

v) As I am not an expert in theoretical calculations, please have the paper reviewed also by an expert from this field.

Author Response

First, we would like to thank this reviewer for his/her careful reading of the manuscript, corrections and suggestions. The changes made are detailed below:

Comments and Suggestions for Authors

The manuscript by Antonio Frontera et al describes synthesis and characterization of three copper(II) coordination compounds, two with phenanthroline and one with Hdmpz ligand. The characterization of compounds is complete and the study in general performed in a thorough manner. Nevertheless, the authors should resolve the following issues prior to publication.

i) The authors should comment upon the choice of ligands, e., phen vs.Hdmpz. Without some kind of explanation, a common point (or a difference) between the phen and the Hdmpz complexes is not obvious.

Reply: Both phen and Hdmpz are the N-donor heterocycles can be effectively used as building blocks to develop coordination compounds of desired dimensionalities and potential applications (https://doi.org/10.1021/acs.inorgchem.9b01424; https://doi.org/10.1002/aoc.6247; https://doi.org/10.1016/j.molstruc.2020.129749). We have now incorporated these points in the revised manuscript.

ii) Compounds 1 and 2 differ in the content of the water molecules of crystallization, apart from that, the coordination species are, at least as far as the content is concerned, the same. When describing the crystal structures of 1 and 2, the authors should focus also on the similarities or differences between the complex species [Cu(phen)2Cl]+ of 1 and 2. Perhaps a drawing with overlaid [Cu(phen)2Cl]+ ions of 1 and 2 should be included.

Reply: We have described the crystal structures together for clarity of the differences and similarities of the compounds in the revised manuscript. Both the compounds exhibit cooperative anion–π/π–π/π–anion assemblies which have been addressed collectively in the crystal packing. The overlay diagram of the compounds has been represented using Crystalcmp software (in ESI).

iii) The description of the supramolecular structures in 13 is too detailed and too lengthy and as such very difficult to follow. I suggest to have it shortened.

Reply: We have modified the description as suggested. Two figure files (Figures 5 and 7) have been also shifted to ESI.

iv) lines 521 and 526

When describing the TG curves of 1 and 2, the authors stated that water molecules were decomposed. Such a formulation is wrong. The compounds underwent the loss of water molecules of crystallization. Please correct.

Reply: We have corrected the sentences as suggested by the esteemed reviewer.

v) As I am not an expert in theoretical calculations, please have the paper reviewed also by an expert from this field.

 Reply: We thank the reviewer for kind review of the manuscript

Reviewer 3 Report

Manuscript reports on synthesis of three mixed-ligand complexes of copper(II), their X-ray single-crystal structures, thermal behavior, FTIR, UV-Vis, and QTAIM analysis. The manuscript in general fits aim and scope of the journal, but it cannot be accepted for publication in present form due to serious methodological errors. In particular, the structural model of compound 1 is wrong.

1. Crystal structure of 1 presented in manuscript contains empty solvent-accessible voids of large volume. This is also confirmed by PLATON/CheckCIF report (attached by authors), see PLAT606_ALERT. Detailed analysis of structure solution performed by a reviewer revealed that authors had applied SQUEEZE procedure (mask procedure in olex2) to modify the experimental intensities and to neglect the contribution of disordered ‘solvent molecules’ which were located in the voids. In general, it is a legal procedure but crystallographer must describe it in the crystallographic section always when apply it.

For the certain structure 1 it is not a case, because it is not a ‘solvent molecule’ but a disordered nitrate-anion which is located inside the ‘void’. Indeed, the reported in CIF moiety formula is ‘2(C24 H16 Cl Cu N4), N O3, 4(H2 O), 2(H1.50 O) [+SOLVENT]’, that means one NO3- anion per two [Cu(phen)2Cl]+ cations. For charge compensation reason one needs one more nitrate anion per two [Cu(phen)2Cl]+ cations, i.e. 4 additional anions per unit cell. This roughly agrees with integrated electron density within the voids, namely 156e observed, 128e calculated for four NO3- anions. Extra electrons within the voids may originate from four additional water molecules.

Anyway, the squeezing of anion is strongly prohibited. This is a serious mistake. Authors are strongly requested to perform careful structure analysis.

It worth noting that, the crystal structure of compound 1 was previously reported at room temperature [DOI: 10.13763/j.cnki.jhebnu.nse.2015.02.009], see point 3 of this revision, and it indeed contains two different nitrate anions per a pair of two [Cu(phen)2Cl]+ cations.

Additionally, disordered NO3- anions localized in the voids participate in H-bond with O1w-H1wb, d (D...A) = 2.8-2.9Å.

2. Additionally to point 1, the reported moiety formula ‘2(C24 H16 Cl Cu N4), N O3,

4(H2 O), 2(H1.50 O) [+SOLVENT] is senseless, since it contains ‘H1.5O’ species. This originated from errors in instruction file. Namely, embedded CIF I found the following fragment in _shelx_res_file section:

O1W 6 0.097916 0.663582 -0.001013 11.00000 0.03969 0.05837 =

0.04181 -0.00801 0.00764 0.01600

H1WA 2 0.136250 0.660463 -0.024663 11.00000 -1.50000

H1WB 2 0.085087 0.637167 0.030512 11.00000 -1.50000

PART -1

H2W 2 0.467871 0.707245 0.965682 10.50000 -1.50000

PART 0

H2WA 2 0.488910 0.664679 0.987454 11.00000 -1.50000

PART 1

H2WB 2 0.486696 0.683267 0.802203 10.50000 -1.50000

PART 0

PART -1

H3W 2 0.479385 0.760330 1.197056 10.50000 -1.50000

PART 0

H3WA 2 0.436860 0.781976 1.055628 10.50000 -1.50000

PART 1

H3WB 2 0.455647 0.736040 1.033733 10.50000 -1.50000

PART 0

O2A 6 0.469929 0.616479 0.360203 11.00000 0.02504 0.03735 =

0.03692 -0.00893 0.00442 0.00996

O3A 6 0.500000 0.556831 0.250000 10.50000 0.02649 0.01964 =

0.04042 0.00000 -0.00232 0.00000

N1A 5 0.500000 0.596334 0.250000 10.50000 0.01353 0.02215 =

0.02483 0.00000 -0.00097 0.00000

O2W 6 0.469449 0.682347 0.907808 11.00000 0.03723 0.03642 =

0.03248 0.00005 0.00443 0.00210

O3W 6 0.454772 0.758426 1.098065 11.00000 0.02500 0.02896 =

0.05041 0.00139 0.00790 -0.00543

Please, note that all three H-atoms of disordered O3w water molecule have SOF equals to 0.5, this resulted in wrong composition H1.5O of water. The correct SOF should be 1 for H3wa according to H-bond analysis.

Additionally, I have to point out that authors used -1.50000 constrain for Uiso of H-atoms incorrectly. Namely, the constrained H-atom must follows its parent atom. In this case, Uiso of H-atoms will be fixed equal to 1.5 times higher that Ueq of parent atom. Obviously, the O2w not O1w is a parent for H2w, H2wa, H2wb; the O3w not O1w is a parent atom for H3w, H3wa and H3wb.

Therefor, the correct fragment of instruction file must be as following:

O1W 6 0.097916 0.663582 -0.001013 11.00000 0.03969 0.05837 =

0.04181 -0.00801 0.00764 0.01600

H1WA 2 0.136250 0.660463 -0.024663 11.00000 -1.50000

H1WB 2 0.085087 0.637167 0.030512 11.00000 -1.50000

O2W 6 0.469449 0.682347 0.907808 11.00000 0.03723 0.03642 =

0.03248 0.00005 0.00443 0.00210

PART -1

H2W 2 0.467871 0.707245 0.965682 10.50000 -1.50000

PART 0

H2WA 2 0.488910 0.664679 0.987454 11.00000 -1.50000

PART 1

H2WB 2 0.486696 0.683267 0.802203 10.50000 -1.50000

PART 0

O3W 6 0.454772 0.758426 1.098065 11.00000 0.02500 0.02896 =

0.05041 0.00139 0.00790 -0.00543

PART -1

H3W 2 0.479385 0.760330 1.197056 10.50000 -1.50000

PART 0

H3WA 2 0.436860 0.781976 1.055628 11.00000 -1.50000

PART 1

H3WB 2 0.455647 0.736040 1.033733 10.50000 -1.50000

PART 0

O2A 6 0.469929 0.616479 0.360203 11.00000 0.02504 0.03735 =

0.03692 -0.00893 0.00442 0.00996

O3A 6 0.500000 0.556831 0.250000 10.50000 0.02649 0.01964 =

0.04042 0.00000 -0.00232 0.00000

N1A 5 0.500000 0.596334 0.250000 10.50000 0.01353 0.02215 =

0.02483 0.00000 -0.00097 0.00000

Corrections of SOF and atom order resulted in decreasing of R-factors from R1=2.89 to R1=2.85%.

3. Crystal structures of all three compounds have been reported earlier by other researchers. Namely, crystal structure of 1 at room temperature was reported by Mengli Li, Qing Shi, Zhen Wei, Huihua Song, Hebei Shifan Daxue Xuebao (Ziran Kexueban), 2015, 39, 138, DOI: 10.13763/j.cnki.jhebnu.nse.2015.02.009 , CCDC refcode UJASOQ, deposition number 726816, R1 = 5.65%

Crystal structure of 2 was reported three times,

- D.Boys, Acta Crystallographica,Section C: Crystal Structure Communications, 1988, 44, 1539, DOI: 10.1107/S0108270188005281 , CCDC refcode GEXXEM, deposition number 1166932

at 295K, with R1 4.30%

- N.Kumari, B.D.Ward, S.Kar, L.Mishra, Polyhedron, 2012, 33, 425, DOI: 10.1016/j.poly.2011.12.009 , CCDC refcode GEXXEM01, deposition number 752079

at 293K, 4.10% (better than in present manuscript!)

- Gautam Gogoi, Jayanta K. Nath, Nazimul Hoque, Subir Biswas, Nand K. Gour, Dhruba Jyoti Kalita, Smiti Rani Bora, Kusum K. Bania, Applied Catalysis A: General, 2022, 644, 118816, DOI: 10.1016/j.apcata.2022.118816, CCDC refcode GEXXEM03, deposition number 2166801

at 296K, 4.64%

Crystal structure 3 was reported by

V.Chandrasekhar, S.Kingsley, A.Vij, K.C.Lam, A.L.Rheingold, Inorganic Chemistry, 2000, 39, 3238, DOI: 10.1021/ic991255k , CCDC refcode WIHQAF, deposition number 148860

at 173K with R1 = 4.55%.

Authors of present manuscript missed that ALL THREE structures had been reported before, and mentioned only the latest paper which reported only structure 3. All previous works must be properly cited in the introduction section.

4. Present manuscript, in its current form, is overblown. Since all the crystal structures are already known their description should be reduced significantly, only novel aspects of intermolecular interaction should be briefly discussed. The comparison of structures 1, 2, and, may be, 3 would be valuable instead of lots C...C and C-H...C distances. ‘Section 3.3.1 FT-IR spectroscopy’ brings nothing to understanding of both compound composition and intermolecular interaction, it is better to transferred it to supplementary closer to respective figure S5.

5. Synthesis of compounds 1-3 was performed due to anion-exchange reaction with CaCl2. In general, exchange reaction is a perfect way to only one product when this product is almost insoluble. This is not a case here, and several by-products e.g. CuCl2*2H2O, CaCl2*2H2O, phen*H2O, are expected as a result of low-temperature crystallization of mother liquor. Synthesis due to e.g. direct interaction of Cu(NO3)2 with CuCl2 in a presence of phen seems to be more rational. What was the reason to use CaCl2? To confirm the phase purity of isolated compounds authors are requested to present powder XRD patterns and compare their with calculated ones.

6. Section 3.4. ‘Thermogravimetric analysis’ describes thermal behavior of compound 1-3 in a very unclear way, e.g. ‘Cl moiety is decomposed’. Please, specify the chemical reaction or eliminated species for every stage of weight loss. Compound 1 shows significant disagreement between observed and calculated weight loss ‘observed weight loss of 8.26% (calcd. = 9.93%)’. This may be due to presence of admixture, therefore the PXRD analysis is required.

7. Section 2.4. ‘The interaction energies of the compounds and supramolecular assemblies investigated in this work were computed at the RI-BP86-D3/def2-TZVP [43-44] level of theory using the experimental geometries and the program Turbomole 7.2’ This is a methodological error. It is well known, that X-H (X=C, N, O, etc.) distances determined by X-ray are always by ca. 0.1Å shorter that correct values determined by neutron diffraction of DFT calculations. This originates from significant polarization of X-H bond and absence of core electrons of H-atoms. Therefore, before any DFT calculations one needs to correct the position of H-atoms at least by shifting then away from corresponding X atoms by ca 0.1Å. In case of intermolecular interactions modeling the correct position of H atoms are crucial since H-atoms are peripheral. Thus, accurate estimation of intermolecular interaction requires DFT optimization of molecular geometry. This is another issue because this optimization must be performed within periodic boundary conditions, otherwise significant part of intermolecular interactions will be neglected. Manuscript does not describe the simulated model (why?), but one can guess (because of Turbomole capabilities) that the calculations were performed for a cluster of several isolated molecules, not a crystal. If the molecular geometries were optimized, the corresponding procedure should be decribed in experimental section in a detail.

8. There are a number of less serious flaws:

- Table 1, empirical formula of compound 1 does not correspond to the one in a text ‘[Cu(phen)2Cl]NO3·3H2O’, this is due to mistake in structural model.

- line 153, ‘D8 Venture diffractometer, ’ should be ‘Bruker D8 Venture diffractometer, ’

- line 158, ‘were solved by direct method’ this contradict the CIF’s where I found that the dual-space phasing by SHELXT was used for structure solution.

- line 162, ‘Hydrogen atoms were inserted at calculated positions and refined as riders ’. Is is right even for H2O molecules?

- Figures 1 and 5 show H3O species which confuses the readers. Indeed, one can understand the H3O as hydroxonium cation, but this is just a disordered water molecule. Please, show the only one position of disordered molecules, the other position may be indicated as transparent of dash line.

- Table 3 shows interatomic distances without uncertainties. The ‘C7‒H7⋯Cl1 2.90 ’ (line 473) seems to be wrong.

- line 514, ‘both solid and in aqueous phases ’ how is it possible? The compound 2 was prepared in methanol, was not it?

- Figures 13 and 14, what are the differences between left and right panels?

- Section 3.5. Please provide the characteristics of BCPs (density, Laplacian, coordinates).

Author Response

First, we would like to thank this reviewers for his/her careful reading of the manuscript, important corrections ans suggestions. The changes made are detailed below:

Manuscript reports on synthesis of three mixed-ligand complexes of copper(II), their X-ray single-crystal structures, thermal behavior, FTIR, UV-Vis, and QTAIM analysis. The manuscript in general fits aim and scope of the journal, but it cannot be accepted for publication in present form due to serious methodological errors. In particular, the structural model of compound 1 is wrong.

Q1. Crystal structure of 1 presented in manuscript contains empty solvent-accessible voids of large volume. This is also confirmed by PLATON/CheckCIF report (attached by authors), see PLAT606_ALERT. Detailed analysis of structure solution performed by a reviewer revealed that authors had applied SQUEEZE procedure (mask procedure in olex2) to modify the experimental intensities and to neglect the contribution of disordered ‘solvent molecules’ which were located in the voids. In general, it is a legal procedure but crystallographer must describe it in the crystallographic section always when apply it.

For the certain structure 1 it is not a case, because it is not a ‘solvent molecule’ but a disordered nitrate-anion which is located inside the ‘void’. Indeed, the reported in CIF moiety formula is ‘2(C24 H16 Cl Cu N4), N O3, 4(H2 O), 2(H1.50 O) [+SOLVENT]’, that means one NO3- anion per two [Cu(phen)2Cl]+ cations. For charge compensation reason one needs one more nitrate anion per two [Cu(phen)2Cl]+ cations, i.e. 4 additional anions per unit cell. This roughly agrees with integrated electron density within the voids, namely 156e observed, 128e calculated for four NO3- anions. Extra electrons within the voids may originate from four additional water molecules.

Anyway, the squeezing of anion is strongly prohibited. This is a serious mistake. Authors are strongly requested to perform careful structure analysis.

 It worth noting that, the crystal structure of compound 1 was previously reported at room temperature [DOI: 10.13763/j.cnki.jhebnu.nse.2015.02.009], see point 3 of this revision, and it indeed contains two different nitrate anions per a pair of two [Cu(phen)2Cl]+ cations.

Additionally, disordered NO3- anions localized in the voids participate in H-bond with O1w-H1wb, d (D...A) = 2.8-2.9Å.

 Q2. Additionally to point 1, the reported moiety formula ‘2(C24 H16 Cl Cu N4), N O3,

4(H2 O), 2(H1.50 O) [+SOLVENT]’ is senseless, since it contains ‘H1.5O’ species. This originated from errors in instruction file. Namely, embedded CIF I found the following fragment in _shelx_res_file section:

 O1W 6 0.097916 0.663582 -0.001013 11.00000 0.03969 0.05837 =

0.04181 -0.00801 0.00764 0.01600

H1WA 2 0.136250 0.660463 -0.024663 11.00000 -1.50000

H1WB 2 0.085087 0.637167 0.030512 11.00000 -1.50000

PART -1

H2W 2 0.467871 0.707245 0.965682 10.50000 -1.50000

PART 0

H2WA 2 0.488910 0.664679 0.987454 11.00000 -1.50000

PART 1

H2WB 2 0.486696 0.683267 0.802203 10.50000 -1.50000

PART 0

PART -1

H3W 2 0.479385 0.760330 1.197056 10.50000 -1.50000

PART 0

H3WA 2 0.436860 0.781976 1.055628 10.50000 -1.50000

PART 1

H3WB 2 0.455647 0.736040 1.033733 10.50000 -1.50000

 PART 0

O2A 6 0.469929 0.616479 0.360203 11.00000 0.02504 0.03735 =

0.03692 -0.00893 0.00442 0.00996

O3A 6 0.500000 0.556831 0.250000 10.50000 0.02649 0.01964 =

0.04042 0.00000 -0.00232 0.00000

N1A 5 0.500000 0.596334 0.250000 10.50000 0.01353 0.02215 =

0.02483 0.00000 -0.00097 0.00000

 O2W 6 0.469449 0.682347 0.907808 11.00000 0.03723 0.03642 =

0.03248 0.00005 0.00443 0.00210

O3W 6 0.454772 0.758426 1.098065 11.00000 0.02500 0.02896 =

0.05041 0.00139 0.00790 -0.00543

Please, note that all three H-atoms of disordered O3w water molecule have SOF equals to 0.5, this resulted in wrong composition H1.5O of water. The correct SOF should be 1 for H3wa according to H-bond analysis.

Reply: When we solved the structure, we decided to solve it using 1 nitrate anion and one hydroxyl anion (in the form of H3O2). We did it like that to have the correct charge compensation. Then we could see additional disordered peaks in the Fourier, which were squeezed. Now, while revising the structure, we have tried to model the disordered peaks, looking for a “neglected” nitrate. Unfortunately, our second analysis did not show a disordered nitrate at all, the peaks are independent, with separations higher than 2 angstroms between them (resembling water molecules). For that reason, we have decided to not modify the refinement.

We have to say that we perfectly understand the referee, it’s forbidden to mask anions. But, we have to say that it’s not our intention at all. We we have not been able to find a better solution. We have even sent out the structure to some colleagues and they consider exactly the same. If the referee has a better way to solve it, we will be every grateful to consider his/her advice.

Additionally, I have to point out that authors used -1.50000 constrain for Uiso of H-atoms incorrectly. Namely, the constrained H-atom must follows its parent atom. In this case, Uiso of H-atoms will be fixed equal to 1.5 times higher that Ueq of parent atom. Obviously, the O2w not O1w is a parent for H2w, H2wa, H2wb; the O3w not O1w is a parent atom for H3w, H3wa and H3wb.

Therefor, the correct fragment of instruction file must be as following:

O1W 6 0.097916 0.663582 -0.001013 11.00000 0.03969 0.05837 =

0.04181 -0.00801 0.00764 0.01600

H1WA 2 0.136250 0.660463 -0.024663 11.00000 -1.50000

H1WB 2 0.085087 0.637167 0.030512 11.00000 -1.50000

O2W 6 0.469449 0.682347 0.907808 11.00000 0.03723 0.03642 =

0.03248 0.00005 0.00443 0.00210

PART -1

H2W 2 0.467871 0.707245 0.965682 10.50000 -1.50000

PART 0

H2WA 2 0.488910 0.664679 0.987454 11.00000 -1.50000

PART 1

H2WB 2 0.486696 0.683267 0.802203 10.50000 -1.50000

PART 0

O3W 6 0.454772 0.758426 1.098065 11.00000 0.02500 0.02896 =

0.05041 0.00139 0.00790 -0.00543

PART -1

H3W 2 0.479385 0.760330 1.197056 10.50000 -1.50000

PART 0

H3WA 2 0.436860 0.781976 1.055628 11.00000 -1.50000

PART 1

H3WB 2 0.455647 0.736040 1.033733 10.50000 -1.50000

PART 0

O2A 6 0.469929 0.616479 0.360203 11.00000 0.02504 0.03735 =

0.03692 -0.00893 0.00442 0.00996

O3A 6 0.500000 0.556831 0.250000 10.50000 0.02649 0.01964 =

0.04042 0.00000 -0.00232 0.00000

N1A 5 0.500000 0.596334 0.250000 10.50000 0.01353 0.02215 =

0.02483 0.00000 -0.00097 0.00000

Reply: Thank you, we had a problem with the automatic sorting of the atoms. It has been corrected

Corrections of SOF and atom order resulted in decreasing of R-factors from R1=2.89 to R1=2.85%.

Q3. Crystal structures of all three compounds have been reported earlier by other researchers. Namely, crystal structure of 1 at room temperature was reported by Mengli Li, Qing Shi, Zhen Wei, Huihua Song, Hebei Shifan Daxue Xuebao (Ziran Kexueban), 2015, 39, 138, DOI: 10.13763/j.cnki.jhebnu.nse.2015.02.009 , CCDC refcode UJASOQ, deposition number 726816, R1 = 5.65%

 Crystal structure of 2 was reported three times,

- D.Boys, Acta Crystallographica,Section C: Crystal Structure Communications, 1988, 44, 1539, DOI: 10.1107/S0108270188005281 , CCDC refcode GEXXEM, deposition number 1166932

at 295K, with R1 4.30%

- N.Kumari, B.D.Ward, S.Kar, L.Mishra, Polyhedron, 2012, 33, 425, DOI: 10.1016/j.poly.2011.12.009 , CCDC refcode GEXXEM01, deposition number 752079

at 293K, 4.10% (better than in present manuscript!)

- Gautam Gogoi, Jayanta K. Nath, Nazimul Hoque, Subir Biswas, Nand K. Gour, Dhruba Jyoti Kalita, Smiti Rani Bora, Kusum K. Bania, Applied Catalysis A: General, 2022, 644, 118816, DOI: 10.1016/j.apcata.2022.118816, CCDC refcode GEXXEM03, deposition number 2166801

at 296K, 4.64%

Crystal structure 3 was reported by

V.Chandrasekhar, S.Kingsley, A.Vij, K.C.Lam, A.L.Rheingold, Inorganic Chemistry, 2000, 39, 3238, DOI: 10.1021/ic991255k , CCDC refcode WIHQAF, deposition number 148860

at 173K with R1 = 4.55%.

Authors of present manuscript missed that ALL THREE structures had been reported before, and mentioned only the latest paper which reported only structure 3. All previous works must be properly cited in the introduction section.

Reply: We have cited all the previous works in the revised manuscript in Results section (Section 3.1). The better crystallographic data of our compounds have been compared (please see ESI).

Q4. Present manuscript, in its current form, is overblown. Since all the crystal structures are already known their description should be reduced significantly, only novel aspects of intermolecular interaction should be briefly discussed. The comparison of structures 1, 2, and, may be, 3 would be valuable instead of lots C...C and C-H...C distances. ‘Section 3.3.1 FT-IR spectroscopy’ brings nothing to understanding of both compound composition and intermolecular interaction, it is better to transferred it to supplementary closer to respective figure S5.

Reply: We have reduced the crystal structure analysis part accordingly (shifted to ESI) and the similarities of compounds 1 and 2 have been described together in the revised manuscript. The C∙∙∙C and C-H∙∙∙C distances have been removed from the text and tabulated in ESI. Section 3.3.1 FT-IR spectroscopy’ has also been shifted to ESI.

Q5. Synthesis of compounds 1-3 was performed due to anion-exchange reaction with CaCl2. In general, exchange reaction is a perfect way to only one product when this product is almost insoluble. This is not a case here, and several by-products e.g. CuCl2*2H2O, CaCl2*2H2O, phen*H2O, are expected as a result of low-temperature crystallization of mother liquor. Synthesis due to e.g. direct interaction of Cu(NO3)2 with CuCl2 in a presence of phen seems to be more rational. What was the reason to use CaCl2? To confirm the phase purity of isolated compounds authors are requested to present powder XRD patterns and compare their with calculated ones.

Reply: Our aim was to incorporate two anions in one compound. We were interested to see the self assembly aspect of molecules/anions in solution and therefore we use CaCl2 for the preparation of all the compounds. However, no powdered products were obtained while stirring the reaction mixtures. Therefore the single crystals obtained were analyzed by single crystal XRD. We have repeated all the reactions, but could not crystallize them again in short time frame from the clear mother liquors.

Also, our institute viz. Cotton University, Guwahati, Assam, India has been recently upgraded to a University and we do not have PXRD instrument. This is for kind consideration of the esteemed reviewer.

Q6. Section 3.4. ‘Thermogravimetric analysis’ describes thermal behavior of compound 1-3 in a very unclear way, e.g. ‘Cl moiety is decomposed’. Please, specify the chemical reaction or eliminated species for every stage of weight loss. Compound 1 shows significant disagreement between observed and calculated weight loss ‘observed weight loss of 8.26% (calcd. = 9.93%)’. This may be due to presence of admixture, therefore the PXRD analysis is required.

Reply: The chemical species eliminated in each step has been represented in the revised manuscript (see ESI). TG analysis of compound 1 has been repeated again and the result is in agreement with the crystal structure.

 Q7. Section 2.4. ‘The interaction energies of the compounds and supramolecular assemblies investigated in this work were computed at the RI-BP86-D3/def2-TZVP [43-44] level of theory using the experimental geometries and the program Turbomole 7.2’ This is a methodological error. It is well known, that X-H (X=C, N, O, etc.) distances determined by X-ray are always by ca. 0.1Å shorter that correct values determined by neutron diffraction of DFT calculations. This originates from significant polarization of X-H bond and absence of core electrons of H-atoms. Therefore, before any DFT calculations one needs to correct the position of H-atoms at least by shifting then away from corresponding X atoms by ca 0.1Å. In case of intermolecular interactions modeling the correct position of H atoms are crucial since H-atoms are peripheral. Thus, accurate estimation of intermolecular interaction requires DFT optimization of molecular geometry. This is another issue because this optimization must be performed within periodic boundary conditions, otherwise significant part of intermolecular interactions will be neglected. Manuscript does not describe the simulated model (why?), but one can guess (because of Turbomole capabilities) that the calculations were performed for a cluster of several isolated molecules, not a crystal. If the molecular geometries were optimized, the corresponding procedure should be decribed in experimental section in a detail.

Reply: The position of the H-atoms has been optimized. This has been clarified in the computational methods. We have not optimized the assemblies because we are interested in analyzing the interactions as they stand in the solid state, thus using the experimental geometry instead of the optimized one. This has been also stated in the computational methods

Q8. There are a number of less serious flaws:

- Table 1, empirical formula of compound 1 does not correspond to the one in a text ‘[Cu(phen)2Cl]NO3·3H2O’, this is due to mistake in structural model.

Reply: Please see the response for Q1 and Q2 for the justification of structural model. The empirical formula of compound 1 in Table 1 has been incorporated as it has been observed in the CIF and checkcif reports.

- line 153, ‘D8 Venture diffractometer, ’ should be ‘Bruker D8 Venture diffractometer, ’

Response: We have modified accordingly.

- line 158, ‘were solved by direct method’ this contradict the CIF’s where I found that the dual-space phasing by SHELXT was used for structure solution.

Response: It has been corrected, structures were solved by intrinsic phasing, using XT.

- line 162, ‘Hydrogen atoms were inserted at calculated positions and refined as riders ’. Is is right even for H2O molecules?

Response: It has been corrected. For compound 1, the H-atoms from water molecules were located in the Fourier maps. For compound 2 H-atoms from water were calculated.

- Figures 1 and 5 show H3O species which confuses the readers. Indeed, one can understand the H3O as hydroxonium cation, but this is just a disordered water molecule. Please, show the only one position of disordered molecules, the other position may be indicated as transparent of dash line.

Response: Figures 1 and 5 have been modified accordingly.

- Table 3 shows inter-atomic distances without uncertainties. The ‘C7‒H7Cl1 2.90 ’ (line 473) seems to be wrong.

Response: Uncertainties are added with the D-A distances. We have corrected the value for C7‒H7⋯Cl1 in the revised manuscript.

- line 514, ‘both solid and in aqueous phases’ how is it possible? The compound 2 was prepared in methanol, was not it?

Response: We have revised the statement in the revised manuscript.

- Figures 13 and 14, what are the differences between left and right panels?

Response: In these Figures (now figures 11 and 12), the right panel corresponds to the mutated dimers. In Figure 11, in the mutated dimer the Cls have been replaced by H-atoms as indicated by the small arrows in Figure 11b. In Figure 12, the mutated dimer has two H-atoms instead of two methyl groups, see also small arrows in Figure 12b.

- Section 3.5. Please provide the characteristics of BCPs (density, Laplacian, coordinates).

Response: This has been done, see new table 3.

Round 2

Reviewer 1 Report

Thank you, all the comments are adressed. The paper can be published after thorous reading and editing misprints.

For example,  geoemtries; noncovalent and non-covalent;  phenmoieties --> phen moieties.

Please see the title: it seems that Nitrate-water should be Nitrate-Water. Please, see that phen is shown in italics in abstract and in page 2 and without it in the other parts of the manuscript. Please, check.

Ref. 32; Kravchenkoa --> Kravchenko; Gippiusb --> Gippius.

Ref. 43: Avdeeva, V.V.; Malinina, E.A.; Yu. K.; Zhizhin, Z.; Kuznetsov, N.T. --> Avdeeva, V.V.; Malinina, E.A.; Zhizhin, K. Yu..; Kuznetsov, N.T.

Author Response

We thank this referee for his/her second reading of the manuscript corrections and suggestions. The changes made and point-by-point responses are detailed below:

Thank you, all the comments are addressed. The paper can be published after thorous reading and editing misprints.

For example, geoemtries; noncovalent and non-covalent;  phenmoieties --> phen moieties.

Please see the title: it seems that Nitrate-water should be Nitrate-Water. Please, see that phen is shown in italics in abstract and in page 2 and without it in the other parts of the manuscript. Please, check.

Ref. 32; Kravchenkoa --> Kravchenko; Gippiusb --> Gippius.

Ref. 43: Avdeeva, V.V.; Malinina, E.A.; Yu. K.; Zhizhin, Z.; Kuznetsov, N.T. --> Avdeeva, V.V.; Malinina, E.A.; Zhizhin, K. Yu..; Kuznetsov, N.T.

 Reply: The aforementioned points have been addressed in the revised manuscript.

Reviewer 3 Report

In revised manuscript authors have resolved only part of questions raised in my revision. The most important one still requires some comments.

Q1. Crystal structure of 1 presented in manuscript contains empty solvent-accessible voids of large volume. This is also confirmed by PLATON/CheckCIF report (attached by authors), see PLAT606_ALERT. Detailed analysis of structure solution performed by a reviewer revealed that authors had applied SQUEEZE procedure (mask procedure in olex2) to modify the experimental intensities and to neglect the contribution of disordered ‘solvent molecules’ which were located in the voids. In general, it is a legal procedure but crystallographer must describe it in the crystallographic section always when apply it.

For the certain structure 1 it is not a case, because it is not a ‘solvent molecule’ but a disordered nitrate-anion which is located inside the ‘void’. Indeed, the reported in CIF moiety formula is ‘2(C24 H16 Cl Cu N4), N O3, 4(H2 O), 2(H1.50 O) [+SOLVENT]’, that means one NO3- anion per two [Cu(phen)2Cl]+ cations. For charge compensation reason one needs one more nitrate anion per two [Cu(phen)2Cl]+ cations, i.e. 4 additional anions per unit cell. This roughly agrees with integrated electron density within the voids, namely 156e observed, 128e calculated for four NO3- anions. Extra electrons within the voids may originate from four additional water molecules.

Anyway, the squeezing of anion is strongly prohibited. This is a serious mistake. Authors are strongly requested to perform careful structure analysis.

It worth noting that, the crystal structure of compound 1 was previously reported at room temperature [DOI: 10.13763/j.cnki.jhebnu.nse.2015.02.009], see point 3 of this revision, and it indeed contains two different nitrate anions per a pair of two [Cu(phen)2Cl]+ cations.

Additionally, disordered NO3- anions localized in the voids participate in H-bond with O1w-H1wb, d (D...A) = 2.8-2.9Å.

Q2. Additionally to point 1, the reported moiety formula ‘2(C24 H16 Cl Cu N4), N O3,

4(H2 O), 2(H1.50 O) [+SOLVENT]’ is senseless, since it contains ‘H1.5O’ species.

Please, note that all three H-atoms of disordered O3w water molecule have SOF equals to 0.5, this resulted in wrong composition H1.5O of water. The correct SOF should be 1 for H3wa according to H-bond analysis.

Reply: When we solved the structure, we decided to solve it using 1 nitrate anion and one hydroxyl anion (in the form of H3O2). We did it like that to have the correct charge compensation. Then we could see additional disordered peaks in the Fourier, which were squeezed. Now, while revising the structure, we have tried to model the disordered peaks, looking for a “neglected” nitrate. Unfortunately, our second analysis did not show a disordered nitrate at all, the peaks are independent, with separations higher than 2 angstroms between them (resembling water molecules). For that reason, we have decided to not modify the refinement.

We have to say that we perfectly understand the referee, it’s forbidden to mask anions. But, we have to say that it’s not our intention at all. We we have not been able to find a better solution. We have even sent out the structure to some colleagues and they consider exactly the same. If the referee has a better way to solve it, we will be every grateful to consider his/her advice.

Additionally, I have to point out that authors used -1.50000 constrain for Uiso of H-atoms incorrectly. Namely, the constrained H-atom must follows its parent atom. In this case, Uiso of H-atoms will be fixed equal to 1.5 times higher that Ueq of parent atom. Obviously, the O2w not O1w is a parent for H2w, H2wa, H2wb; the O3w not O1w is a parent atom for H3w, H3wa and H3wb.

Reply: Thank you, we had a problem with the automatic sorting of the atoms. It has been corrected

Comment_to_r2: The authors insist that their solution of the crystal structure does not require corrections. However, the deciphered crystal structure of 1 does not correspond to the formula of compound 1 in the text of the article. Indeed, in the text, lines 12, 88, 120, 234 and the scheme, the formula [Cu(phen)2Cl](NO3) 3H2O is the same as C24H22ClCuN5O6 or, based on two copper atoms, C48H44Cl2Cu2N10O12 = [Cu(phen)2Cl]2(NO3)2(H2O)6.
At the same time, the structure model proposed by the authors corresponds to the formula C48H43Cl2Cu2N9O9.
This formula is listed in CIF and in Table 1. According to the authors’ response formula is for [Cu(phen)2Cl]2(NO3)(OH)(H2O)5. Therefore, this formula differs from reported in manuscript for compound 1 by one (NO3)- anion and one H-atom (OH instead of H2O).

It worth noting that this confusion with formulas appears in CHN analysis lines 120 and 128 report different composition for compound 1.

So, where is the correct composition?

Concerning structure solutions. I suggested that NO3- anion is located somewhere in the void and performed refinement against unsqueezed (unmasked) HKL dataset. It worth noting that dataset has poor data-to-parameter ration (ca 11) and anion is disordered at least over two positions within the void with partial occupancy. Thus it accurate description requires more careful analysis than I did. Nevertheless, the results are as follows:

Unsqueezed dataset with no extra species within the void gives R1 = 7.5% and residual density peak of 3.9 e.

The same dataset with additional 0.5 NO3- and 0.5 O (additional H2O molecule was expected according to integrated number of electrons, see above) gives R1 = 4.5% and residual density peak of 1.8e. Please, find the attached  draft.

Certainly, this is R1 is much higher than for squeezed dataset, but squeeze procedure is not applicable in this case. Perhaps more detailed analysis would reveal additional position for disordered species and resulted R-factors would be lower.

Q5. Synthesis of compounds 1-3 was performed due to anion-exchange reaction with CaCl2. In general, exchange reaction is a perfect way to only one product when this product is almost insoluble. This is not a case here, and several by-products e.g. CuCl2*2H2O, CaCl2*2H2O, phen*H2O, are expected as a result of low-temperature crystallization of mother liquor. Synthesis due to e.g. direct interaction of Cu(NO3)2 with CuCl2 in a presence of phen seems to be more rational. What was the reason to use CaCl2? To confirm the phase purity of isolated compounds authors are requested to present powder XRD patterns and compare their with calculated ones.

Reply: Our aim was to incorporate two anions in one compound. We were interested to see the self assembly aspect of molecules/anions in solution and therefore we use CaCl2 for the preparation of all the compounds. However, no powdered products were obtained while stirring the reaction mixtures. Therefore the single crystals obtained were analyzed by single crystal XRD. We have repeated all the reactions, but could not crystallize them again in short time frame from the clear mother liquors.

Also, our institute viz. Cotton University, Guwahati, Assam, India has been recently upgraded to a University and we do not have PXRD instrument. This is for kind consideration of the esteemed reviewer.

Comment_to_r2: Authors have reported the yields for three compounds to be as high as 0.5-0.9 g this is much more than several single-crystals and it is enough for CHN, FTIR and PXRD analysis as well.

Concerning PXRD data without PXRD diffractometer, it is possible to collect high-quality powder XRD pattern in Debye-Scherrer-like geometry using single-crystal diffractometer with 2D detector. Please, consult with book by Bob He (He, Bob B., Two-dimensional x-ray diffraction / by Bob Baoping He. Second edition. Hoboken, NJ : John Wiley & Sons, 2018. ) and Bruker webinars. Note, the recent paper https://doi.org/10.1107/S1600576722005878 describes PXRD data collection procedure using Bruker PhotonIII C14 detector.

Q6. Section 3.4. ‘Thermogravimetric analysis’ describes thermal behavior of compound 1-3 in a very unclear way, e.g. ‘Cl moiety is decomposed’. Please, specify the chemical reaction or eliminated species for every stage of weight loss. Compound 1 shows significant disagreement between observed and calculated weight loss ‘observed weight loss of 8.26% (calcd. = 9.93%)’. This may be due to presence of admixture, therefore the PXRD analysis is required.

Reply: The chemical species eliminated in each step has been represented in the revised manuscript (see ESI). TG analysis of compound 1 has been repeated again and the result is in agreement with the crystal structure.

Comment_to_r2: According to Figure S13 decomposition of [Cu(phen)2Cl](NO3) formed from 1 after 240oC resulted in formation of [Cu(phen)2](NO3). Does this product contain copper(i)?

Why the thermal behavior of [Cu(phen)2Cl](NO3) formed from 1 and 2 are different?

Author Response

We want to thank this referee again for his/her second reading of the manuscript, corrections and suggestions. We have decided to eliminate completely compound 1 from the manuscript to avoid any mistake that could be derived from the wrong interpretation of the X-ray data. Our responses follow:

In revised manuscript authors have resolved only part of questions raised in my revision. The most important one still requires some comments.

Q1. Crystal structure of 1 presented in manuscript contains empty solvent-accessible voids of large volume. This is also confirmed by PLATON/CheckCIF report (attached by authors), see PLAT606_ALERT. Detailed analysis of structure solution performed by a reviewer revealed that authors had applied SQUEEZE procedure (mask procedure in olex2) to modify the experimental intensities and to neglect the contribution of disordered ‘solvent molecules’ which were located in the voids. In general, it is a legal procedure but crystallographer must describe it in the crystallographic section always when apply it.

For the certain structure 1 it is not a case, because it is not a ‘solvent molecule’ but a disordered nitrate-anion which is located inside the ‘void’. Indeed, the reported in CIF moiety formula is ‘2(C24 H16 Cl Cu N4), N O3, 4(H2 O), 2(H1.50 O) [+SOLVENT]’, that means one NO3- anion per two [Cu(phen)2Cl]+ cations. For charge compensation reason one needs one more nitrate anion per two [Cu(phen)2Cl]+ cations, i.e. 4 additional anions per unit cell. This roughly agrees with integrated electron density within the voids, namely 156e observed, 128e calculated for four NO3- anions. Extra electrons within the voids may originate from four additional water molecules.

Anyway, the squeezing of anion is strongly prohibited. This is a serious mistake. Authors are strongly requested to perform careful structure analysis.

It worth noting that, the crystal structure of compound 1 was previously reported at room temperature [DOI: 10.13763/j.cnki.jhebnu.nse.2015.02.009], see point 3 of this revision, and it indeed contains two different nitrate anions per a pair of two [Cu(phen)2Cl]+ cations.

Additionally, disordered NO3- anions localized in the voids participate in H-bond with O1w-H1wb, d (D...A) = 2.8-2.9Å.

Q2. Additionally to point 1, the reported moiety formula ‘2(C24 H16 Cl Cu N4), N O3,

4(H2 O), 2(H1.50 O) [+SOLVENT]’ is senseless, since it contains ‘H1.5O’ species.

Please, note that all three H-atoms of disordered O3w water molecule have SOF equals to 0.5, this resulted in wrong composition H1.5O of water. The correct SOF should be 1 for H3wa according to H-bond analysis.

Reply: When we solved the structure, we decided to solve it using 1 nitrate anion and one hydroxyl anion (in the form of H3O2). We did it like that to have the correct charge compensation. Then we could see additional disordered peaks in the Fourier, which were squeezed. Now, while revising the structure, we have tried to model the disordered peaks, looking for a “neglected” nitrate. Unfortunately, our second analysis did not show a disordered nitrate at all, the peaks are independent, with separations higher than 2 angstroms between them (resembling water molecules). For that reason, we have decided to not modify the refinement.

We have to say that we perfectly understand the referee, it’s forbidden to mask anions. But, we have to say that it’s not our intention at all. We we have not been able to find a better solution. We have even sent out the structure to some colleagues and they consider exactly the same. If the referee has a better way to solve it, we will be every grateful to consider his/her advice.

Additionally, I have to point out that authors used -1.50000 constrain for Uiso of H-atoms incorrectly. Namely, the constrained H-atom must follows its parent atom. In this case, Uiso of H-atoms will be fixed equal to 1.5 times higher that Ueq of parent atom. Obviously, the O2w not O1w is a parent for H2w, H2wa, H2wb; the O3w not O1w is a parent atom for H3w, H3wa and H3wb.

Reply: Thank you, we had a problem with the automatic sorting of the atoms. It has been corrected

Comment_to_r2: The authors insist that their solution of the crystal structure does not require corrections. However, the deciphered crystal structure of 1 does not correspond to the formula of compound 1 in the text of the article. Indeed, in the text, lines 12, 88, 120, 234 and the scheme, the formula [Cu(phen)2Cl](NO3) 3H2O is the same as C24H22ClCuN5O6 or, based on two copper atoms, C48H44Cl2Cu2N10O12 = [Cu(phen)2Cl]2(NO3)2(H2O)6.
At the same time, the structure model proposed by the authors corresponds to the formula C48H43Cl2Cu2N9O9. This formula is listed in CIF and in Table 1. According to the authors’ response formula is for [Cu(phen)2Cl]2(NO3)(OH)(H2O)5. Therefore, this formula differs from reported in manuscript for compound 1 by one (NO3)- anion and one H-atom (OH instead of H2O).

It worth noting that this confusion with formulas appears in CHN analysis lines 120 and 128 report different composition for compound 1.

So, where is the correct composition?

Concerning structure solutions. I suggested that NO3- anion is located somewhere in the void and performed refinement against unsqueezed (unmasked) HKL dataset. It worth noting that dataset has poor data-to-parameter ration (ca 11) and anion is disordered at least over two positions within the void with partial occupancy. Thus it accurate description requires more careful analysis than I did. Nevertheless, the results are as follows:

Unsqueezed dataset with no extra species within the void gives R1 = 7.5% and residual density peak of 3.9 e.

The same dataset with additional 0.5 NO3- and 0.5 O (additional H2O molecule was expected according to integrated number of electrons, see above) gives R1 = 4.5% and residual density peak of 1.8e. Please, find the attached  draft.

Certainly, this is R1 is much higher than for squeezed dataset, but squeeze procedure is not applicable in this case. Perhaps more detailed analysis would reveal additional position for disordered species and resulted R-factors would be lower.

Reply: We have removed compound 1 from the manuscript.

Q5. Synthesis of compounds 1-3 was performed due to anion-exchange reaction with CaCl2. In general, exchange reaction is a perfect way to only one product when this product is almost insoluble. This is not a case here, and several by-products e.g. CuCl2*2H2O, CaCl2*2H2O, phen*H2O, are expected as a result of low-temperature crystallization of mother liquor. Synthesis due to e.g. direct interaction of Cu(NO3)2 with CuCl2 in a presence of phen seems to be more rational. What was the reason to use CaCl2? To confirm the phase purity of isolated compounds authors are requested to present powder XRD patterns and compare their with calculated ones.

Reply: Our aim was to incorporate two anions in one compound. We were interested to see the self assembly aspect of molecules/anions in solution and therefore we use CaCl2 for the preparation of all the compounds. However, no powdered products were obtained while stirring the reaction mixtures. Therefore the single crystals obtained were analyzed by single crystal XRD. We have repeated all the reactions, but could not crystallize them again in short time frame from the clear mother liquors.

Also, our institute viz. Cotton University, Guwahati, Assam, India has been recently upgraded to a University and we do not have PXRD instrument. This is for kind consideration of the esteemed reviewer.

Comment_to_r2: Authors have reported the yields for three compounds to be as high as 0.5-0.9 g this is much more than several single-crystals and it is enough for CHN, FTIR and PXRD analysis as well.

Concerning PXRD data without PXRD diffractometer, it is possible to collect high-quality powder XRD pattern in Debye-Scherrer-like geometry using single-crystal diffractometer with 2D detector. Please, consult with book by Bob He (He, Bob B., Two-dimensional x-ray diffraction / by Bob Baoping He. Second edition. Hoboken, NJ : John Wiley & Sons, 2018. ) and Bruker webinars. Note, the recent paper https://doi.org/10.1107/S1600576722005878 describes PXRD data collection procedure using Bruker PhotonIII C14 detector.

Reply: We replied this query during revision round 1. Please note that we could not crystallize the compound again.

Q6. Section 3.4. ‘Thermogravimetric analysis’ describes thermal behavior of compound 1-3 in a very unclear way, e.g. ‘Cl moiety is decomposed’. Please, specify the chemical reaction or eliminated species for every stage of weight loss. Compound 1 shows significant disagreement between observed and calculated weight loss ‘observed weight loss of 8.26% (calcd. = 9.93%)’. This may be due to presence of admixture, therefore the PXRD analysis is required.

Reply: The chemical species eliminated in each step has been represented in the revised manuscript (see ESI). TG analysis of compound 1 has been repeated again and the result is in agreement with the crystal structure.

Comment_to_r2: According to Figure S13 decomposition of [Cu(phen)2Cl](NO3) formed from 1 after 240oC resulted in formation of [Cu(phen)2](NO3). Does this product contain copper(i)?

Reply: Compound 1 has been now removed from the manuscript.

Why the thermal behavior of [Cu(phen)2Cl](NO3) formed from 1 and 2 are different?
Resply: Compound 1 has been now removed from the manuscript.

Round 3

Reviewer 3 Report

Authors excluded the problematic structure from the manuscript. Altought the manuscript has lost some of the benefits of comparing related compounds (former 1 and 2) the revised manuscript does not contain methodoligical flaws and can be accepted.